# Cysteines 1078 and 2991 cross-linking plays a critical role in redox regulation of cardiac ryanodine receptor (RyR)

Roman Nikolaienko [1], Elisa Bovo [1], Daniel Kahn [1], Ryan Gracia[1], Thomas Jamrozik [1] & Aleksey V. Zima [1] ✉

The most common cardiac pathologies, such as myocardial infarction and heart failure, are associated with oxidative stress. Oxidation of the cardiac ryanodine receptor (RyR2) $Ca^{2+}$ channel causes spontaneous oscillations of intracellular $Ca^{2+}$, resulting in contractile dysfunction and arrhythmias. RyR2 oxidation promotes the formation of disulfide bonds between two cysteines on neighboring RyR2 subunits, known as intersubunit cross-linking. However, the large number of cysteines in RyR2 has been a major hurdle in identifying the specific cysteines involved in this pathology-linked post-translational modification of the channel. Through mutagenesis of human RyR2 and in-cell $Ca^{2+}$ imaging, we identify that only two cysteines (out of 89) in each RyR2 subunit are responsible for half of the channel's functional response to oxidative stress. Our results identify cysteines 1078 and 2991 as a redox-sensitive pair that forms an intersubunit disulfide bond between neighboring RyR2 subunits during oxidative stress, resulting in a pathological "leaky" RyR2 $Ca^{2+}$ channel.

Calcium ion ($Ca^{2+}$) is a universal second messenger that regulates a myriad of cellular processes, including contraction, gene regulation, synaptic transmission, hormone secretion, migration, and cell growth[1]. In cardiac myocytes, $Ca^{2+}$ released from the sarcoplasmic reticulum (SR) through the type 2 ryanodine receptor (RyR2) is essential for initiating a robust myocardial contraction[2]. Consequently, defects in RyR2 regulation cause contractile dysfunction and decreased cardiac output. The most common cardiac pathologies with fatal outcomes, such as myocardial infarction and heart failure, are associated with abnormal $Ca^{2+}$ regulation[3]. Pathological alterations during those conditions commonly involve substantial structural and functional remodeling of RyR2 by oxidative stress[4–7]. It has been shown that reactive oxygen species (ROS) can directly activate RyR2-mediated $Ca^{2+}$ release in the form of diastolic spontaneous $Ca^{2+}$ sparks[8,9] and $Ca^{2+}$ waves[4,10], causing arrhythmias and depletion of SR $Ca^{2+}$ load. Oxidative stress also causes the dissociation of an important regulator of RyR2 function, calmodulin (CaM)[11–13]. As a result, RyR2 oxidation has a significant impact on $Ca^{2+}$ signaling and contraction in infarcted and failing hearts[4,6,14–16].

Because RyR2 comprises up to 89 cysteines per subunit[17], it is no surprise that RyR2 plays a key role in the cellular response to oxidative stress. However, a large number of cysteines in RyR2 has been a major limitation in identifying the specific sites of ROS action. As a result, the current understanding of the molecular mechanisms of redox regulation and redox-sensitive sites within RyR2 remains incomplete. Hence, the prospect of targeted therapy against RyR2 oxidation is absent. In contrast, functionally important phosphorylation sites on RyR2 have been identified and characterized[18–21], and pre-clinical studies targeting these sites are currently underway. Thus, identifying the functionally important cysteines in RyR2 is essential for advancing our understanding of molecular mechanisms of RyR2 dysfunction during pathological conditions associated with oxidative stress.

In our previous studies, we discovered a novel posttranslational modification (PTM) of RyR2 induced by oxidative stress. As a common mechanism of channel assembly, the RyR2 complex is formed by four identical subunits. We have shown that oxidative stress can promote the formation of disulfide bonds between cysteines on neighboring

[1]Department of Cell and Molecular Physiology, Loyola University Chicago, Maywood, IL 60153, USA. ✉e-mail: azima@luc.edu

RyR2 subunits, also known as intersubunit cross-linking[13,22]. There is structural evidence that a transition of RyR between the closed and the opened state (i.e., channel gating) is associated with an intersubunit dynamic[23]. Thus, we hypothesized that intersubunit cross-linking should significantly impact RyR2 function and $Ca^{2+}$ signaling. In agreement with our hypothesis, we found that in ventricular myocytes from healthy hearts, the degree of SR $Ca^{2+}$ leak induced by oxidative stress positively correlated with the level of RyR2 cross-linking[13,22]. Moreover, a significant level of RyR2 cross-linking has been detected in ventricular myocytes from failing hearts[7].

The effect of oxidative stress on the channel's structure and function is better described for the skeletal muscle type 1 RyR isoform (RyR1). Historically, the oxidation of thiol groups in RyR1 by reactive sulfhydryl agents has been linked to increased SR $Ca^{2+}$ release in skeletal muscle fibers[24]. It has been suggested that cysteine 3635 in RyR1 is the most important residue responsible for redox regulation of the channel[25,26]. It has been shown that cysteine 3635 can be oxidized during oxidative stress to form a disulfide bond[27,28]. We re-evaluated the functional significance of the corresponding cysteine 3602 in RyR2 and showed that this residue is not involved in channel cross-linking nor $Ca^{2+}$ leak activation by oxidative stress[13]. It appears that despite ~70% homology between RyR1 and RyR2, these channels exhibit important structural and functional differences, particularly in redox regulation. In the end, the specific mechanisms of redox regulation and the redox-sensitive residues within RyR2 remain largely unknown.

In this study, we identify cysteines 1078 and 2991 as a critical redox-sensitive pair that can form a disulfide bond between neighboring RyR2 subunits during oxidative stress, resulting in "leaky" RyR2.

## Results

### Cell system to study the functional role of RyR2 intersubunit cross-linking in $Ca^{2+}$ signaling

To define the functional role of cysteines involved in RyR2 intersubunit cross-linking, we took advantage of our assay to study RyR2 function in living cells[29]. Changes in the ER luminal $[Ca^{2+}]$ ($[Ca^{2+}]_{ER}$) were measured with the genetically encoded ER $Ca^{2+}$ sensor R-CEPIA1er in HEK293 cells expressing human RyR2 (hRyR2; Fig. 1a). R-CEPIA1er has an appropriate affinity to measure $[Ca]_{ER}$ dynamics in non-muscle[30] and muscle cells[31]. The cells were also transfected with SERCA2a $Ca^{2+}$ pump to maintain physiological ER $Ca^{2+}$ load. Co-expression of hRyR2 and SERCA2a produced regular $[Ca^{2+}]_{ER}$ depletions (i.e., $Ca^{2+}$ waves) due to spontaneous hRyR2 activation followed by SERCA-mediated $Ca^{2+}$ reuptake (Fig. 1b). Thus, the expression of both hRyR2 and SERCA2a creates a heterologous cell system that reproduces key aspects of cardiac $Ca^{2+}$ signaling. As expected, these $Ca^{2+}$ waves were sensitive to the RyR agonist caffeine. By activating hRyR2, caffeine (10 mM) caused a global ER $Ca^{2+}$ release and $[Ca^{2+}]_{ER}$ depletion. The completeness of $[Ca^{2+}]_{ER}$ depletion by caffeine was validated by the SERCA inhibitor thapsigargin (10 μM). Since the application of thapsigargin did not cause additional $[Ca^{2+}]_{ER}$ depletion, we used the caffeine output as $F_{min}$ during signal normalization (Supplementary Fig. 1). Since HEK293 cells lack endogenous RyR (verified by Western blot analysis and the lack of response to caffeine; Supplementary Fig. 2), this cell system allows studying only recombinant RyR. Thus, we consider this minimalistic $Ca^{2+}$ handling system to be ideal for examining the protective properties of different cysteine mutations against hRyR2 intersubunit cross-linking.

To identify the levels of oxidative stress sufficient to induce hRyR2 cross-linking in our cell model system, we treated cells expressing the wild-type hRyR2 (hRyR2$^{WT}$) with increasing concentrations of the thiol oxidant diamide. Western blot analysis revealed that increasing concentrations of diamide caused the disappearance of the hRyR2$^{WT}$ 560 kDa monomeric band and promoted the formation of cross-linked oligomers. This effect could be reversed by treating cell samples with the reducing agent DTT (Fig. 1c). Next, we tested the effect of diamide on $[Ca^{2+}]_{ER}$ dynamics in our cell model system. For this set of

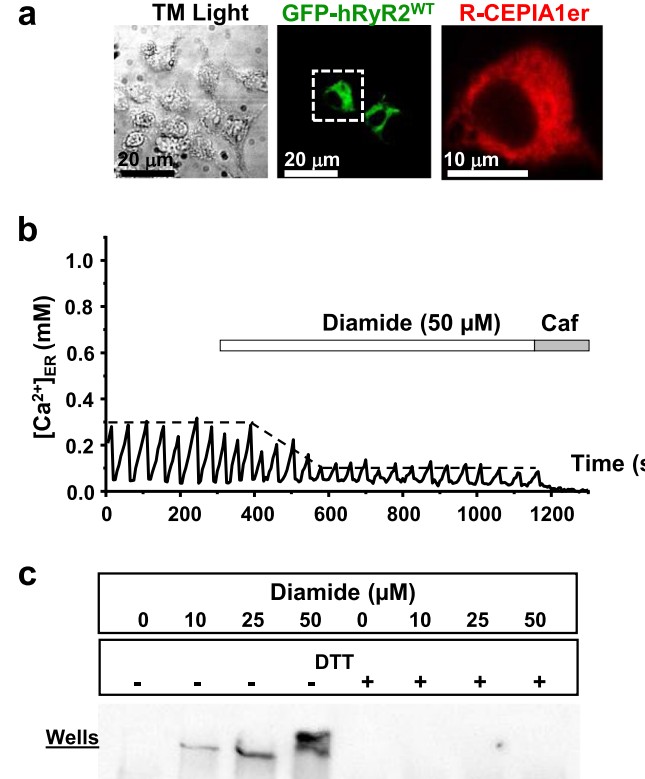

**Fig. 1 | Cell model for functional study of redox-sensitive cysteine residues in hRyR2. a** Representative confocal image of the HEK293 cell model for the functional studies of redox-sensitive cysteines in RyR2. Corresponding images demonstrate the cells under transmitted light and the expression of GFP-tagged hRyR2 and R-CEPIA1er $[Ca^{2+}]_{ER}$ sensor. The images of co-localized GFP-tagged RyR2 and R-CEPIA1er were obtained for corresponding HEK293 cells in all confocal experiments mentioned in this paper. **b** Representative recordings of $Ca^{2+}$ waves generated by hRyR2 in HEK293 cells. Diamide was added to induce hRyR2 oxidation and cross-linking. Caffeine (Caf; 10 mM) and ionomycin (2 μM) in 10 mM $Ca^{2+}$ were used to normalize the signal. The dashed line indicates the level of ER $Ca^{2+}$ load. **c** Western blot of the disulfide cross-linking of hRyR2 caused by increasing concentrations of diamide. DTT (100 mM) was added to the samples to reverse hRyR2 oxidation and cross-linking (data representative of 3 independent experiments).

experiments, we took advantage of stable inducible Flp-In T-Rex-293 cells expressing SERCA2a[29]. We used normalized and averaged values of $Ca^{2+}$ wave peaks (marked by dashed lines; Fig. 1b) to estimate the relative change in $[Ca^{2+}]_{ER}$ loading upon the application of diamide. The concentration of diamide (50 μM) that caused the maximal cross-linking (91.8 ± 1.7%; $n = 13$) also resulted in 50.1 ± 1.4% ($n = 72$) depletion of $[Ca^{2+}]_{ER}$ (or ER $Ca^{2+}$ load) in cells that express hRyR2$^{WT}$ tagged with GFP (Fig. 1a, b). Diamide (50 μM) did not affect $[Ca^{2+}]_{ER}$ in cells that do not express hRyR2$^{WT}$ (Supplementary Fig. 2), suggesting that this level of oxidative stress does not affect other endogenous ion channels/pumps in our cell system. Because SERCA function is unaffected by these diamide concentrations[29], the decline in $[Ca^{2+}]_{ER}$ was mainly mediated by an increase in ER $Ca^{2+}$ leak via hRyR2$^{WT}$.

### N-terminal truncation of hRyR2 domains A and B prevents hRyR2 intersubunit cross-linking and activation of hRyR2 by diamide

To identify hRyR2 cysteine oxidation which promotes cross-linking and ER $Ca^{2+}$ leak, we analyzed the channel's N-terminal region (NTR) structure (residues 1-547), which includes three domains (A, B, and C)[32]. The NTR of RyR2 has been shown to play an important role in

the channel's tetramer formation and regulation of the channel's function[33–36]. It has also been characterized as a "hotspot" for the hRyR2 mutations associated with catecholaminergic polymorphic ventricular tachycardia (CPVT)[37,38], suggesting its important role in channel gating. Based on the analysis of the NTR domains orientation in the available RyR2 cryo-EM models[23,39,40], we decided to focus on domains A and B (residues 1-409), which harbor 9 of 10 NTR cysteines (Fig. 2a). We did not evaluate the role of cysteine 501 from domain C because of its distant location from the intersubunit interface of RyR2.

To study the role of NTR cysteines in hRyR2 cross-linking and hRyR2 redox regulation, we removed domains A and B, generating the hRyR2$^{\Delta AB}$ construct (Fig. 2b). We compared cross-linking levels in hRyR2$^{WT}$, and hRyR2$^{\Delta AB}$ expressed in HEK293 cells after a treatment of cells with diamide. The mutant channel was expressed at a similar level as hRyR2$^{WT}$ (estimated from an intensity of "Diamide 0" bands; Fig. 3a). The truncation of hRyR2 domains A and B significantly prevented cross-linking induced by 25 and 50 μM diamide (Fig. 3a). Overall, the mutation preserved 45.6 ± 7.9% ($n = 4$, $p = 5.2 \times 10^{-4}$) and 43.5 ± 10.7% ($n = 4$, $p = 0.014$) of hRyR2$^{\Delta AB}$ monomers from cross-linking at 25 and 50 μM diamide, correspondingly (Fig. 3b). In these experiments and hereafter, the protective effect of hRyR2 mutations was estimated based on the amount of the preserved monomer, which was calculated as a difference between normalized amounts of the mutant and hRyR2$^{WT}$ monomeric forms.

Next, we evaluated the functional response of hRyR2$^{\Delta AB}$ to diamide in our cell system. It has been previously shown that the truncation of the domains A and B does not cause the channel loss-of-function[34]. While hRyR2$^{\Delta AB}$ mutant in our model responded to caffeine activation (Fig. 3c), this mutation decreased Ca$^{2+}$ wave frequency and increased ER Ca$^{2+}$ load, suggesting an inhibition of hRyR2 activity. The application of 50 μM diamide resulted in 52.7 ± 2.6% ($n = 22$) decline in ER Ca$^{2+}$ load caused by increased ER Ca$^{2+}$ leak through hRyR2$^{WT}$. However, in cells expressing hRyR2$^{\Delta AB}$, the same diamide concentration caused only 11.1 ± 1.3% depletion of [Ca$^{2+}$]$_{ER}$, yielding a relative protective effect of 79% ($n = 20$, $p = 5.2 \times 10^{-17}$, Fig. 3d). Thus, the removal of domains A and B was sufficient to eliminate the most of excessive ER Ca$^{2+}$ leak mediated by hRyR2 oxidation, suggesting a critical role of hRyR2 NTR domains in the channel's activation by oxidative stress.

### Individual cysteines of the NTR of hRyR2 are not directly involved in hRyR2 intersubunit cross-linking

To narrow the search of the cross-linking sites within the NTR of hRyR2, we produced a construct lacking only the A domain (hRyR2$^{\Delta A}$). hRyR2$^{\Delta A}$ showed substantial resistance to cross-linking similar to the one observed for the hRyR2$^{\Delta AB}$ construct (Fig. 4a), preserving 43.8 ± 7.5% ($n = 5$, $p = 2.4 \times 10^{-4}$) and 21.8 ± 8.3% ($n = 5$, $p = 0.0035$) of the hRyR2$^{\Delta A}$ monomeric form after the corresponding treatment with 25 and 50 μM diamide (Fig. 4b). Thus, removing the NTR domain A was sufficient to significantly prevent cross-linking of the channel. Next, we investigated which cysteine residues in domain A are involved in forming intersubunit disulfide bonds during channel oxidation. It has been shown that domain A contains 7 cysteines, and some of them can be partially exposed to solvent and avato oxidation (e.g., cysteines 36 and 158)[17]. However, when we generated a series of hRyR2 constructs with the cysteine to serine mutations in domain A (hRyR2$^{C24S}$ hRyR2$^{C36S}$, hRyR2$^{C47S/C65S}$, hRyR2$^{C131S/C132S/C158S}$), none of them prevented the channel's cross-linking (Supplementary Fig. 3). These results concluded that the individual cysteines in the A domain were not directly involved in hRyR2 cross-linking. Based on the available cryo-EM structural models[23,39–41], cysteines from domain B of hRyR2 could only be cross-linked with cysteines from domain A of the neighboring subunit. Thus, these results also excluded the possibility of the involvement of cysteines in the hRyR2 domain B. It appears that the domains A and B allosterically modulate intersubunit cross-linking of

hRyR2 instead of directly providing thiol groups to form intersubunit disulfide bonds.

### Cysteine 1078 is directly involved in hRyR2 intersubunit cross-linking and activation of hRyR2 by diamide

Since the NTR cysteines are not directly involved in intersubunit cross-linking, we sought to explore other possible hRyR2 regions capable of forming intersubunit disulfide bonds. The recently solved Cryo-EM models of hRyR2[40,41] made it possible to highlight a potential interface for the disulfide bond formation between cysteine 2991 on the helical domain 2 (HD2) loop on one RyR2 subunit and cysteine 1078 of the P1-SPRY2 linker on another subunit (Fig. 5 and Supplementary Fig. 4). To test whether these cysteines are involved in cross-linking, we first designed the construct with cysteine 1078 mutated to serine (hRyR2$^{C1078S}$). Western blot analysis revealed that the hRyR2$^{C1078S}$ mutation substantially prevented cross-linking caused by diamide (Fig. 6a, b). Overall, the mutation protected 58.7 ± 9.4% ($n = 5$, $p = 1.5 \times 10^{-4}$) of hRyR2$^{C1078S}$ monomers from cross-linking at 25 μM and 53.4 ± 7.4% ($n = 5$, $p = 3.2 \times 10^{-5}$) at 50 μM diamide (Fig. 6b).

Our recent data demonstrated that RyR2 intersubunit cross-linking positively correlated with RyR2 activation by oxidative stress[7,13]. We tested whether this residue is essential for the channel's functional response to oxidative stress. The hRyR2$^{C1078S}$ mutant generated Ca$^{2+}$ waves with similar properties as in cells expressing hRyR2$^{WT}$ (Fig. 6c and Supplementary Fig. 5). The application of 25 μM diamide caused the 42.1 ± 2.1% ($n = 43$) decline of ER Ca$^{2+}$ load in cells expressing hRyR2$^{WT}$ and 24.3 ± 2.0% ($n = 40$) in cells expressing hRyR2$^{C1078S}$ (Fig. 6d). Since we observed some decline in hRyR2$^{C1078S}$ monomer levels with an increase in diamide concentration from 25 to 50 μM, we studied the effect of 50 μM on [Ca$^{2+}$]$_{ER}$. Diamide (50 μM) caused the 46.9 ± 2.0% decline ($n = 27$) of ER Ca$^{2+}$ load for hRyR2$^{WT}$ and 28.5 ± 2.0% ($n = 36$) for hRyR2$^{C1078S}$ (Fig. 6d). Thus, the relative protective effect of the C1078S mutation was ~40% for both diamide concentrations (corresponding $p = 2.7 \times 10^{-8}$ and $p = 2.8 \times 10^{-8}$). Since the higher diamide concentration only slightly increased the ER Ca$^{2+}$ load depletion, these results suggest that the most functionally significant disulfide cross-linking occurs at 25 μM diamide (by oxidizing cysteine 1078), while higher diamide concentrations oxidize other hRyR2 cysteines without a substantial increase in ER Ca$^{2+}$ leak.

### Cysteines 1078 and C2991 form the functionally important intersubunit disulfide bond within the hRyR2 complex

To explore whether cysteine 1078 forms the intersubunit disulfide bond with cysteine 2991, we mutated cysteine 2991 to serine (hRyR2$^{C2991S}$). Western blot analysis showed that the C2991S mutation significantly prevented hRyR2 cross-linking, protecting 48.7 ± 7.3% ($n = 3$, $p = 0.0037$) of its monomeric form at 25 μM diamide and 72.3 ± 2.3% ($n = 3$, $p = 2.0 \times 10^{-4}$) at 50 μM diamide (Fig. 7a, b). We next tested whether hRyR2$^{C2991S}$ would have a similar degree of functional protection as seen for hRyR2$^{C1078S}$. Cells expressing the hRyR2$^{C2991S}$ mutant generated Ca$^{2+}$ waves similar to those observed in cells expressing hRyR2$^{WT}$ or hRyR2$^{C1078S}$ (Fig. 7c and Supplementary Fig. 5). The treatment with 25 μM diamide produced 46.1 ± 2.5% ($n = 29$) depletion of ER Ca$^{2+}$ load for hRyR2$^{WT}$ and 25.0 ± 2.8% ($n = 28$) for hRyR2$^{C2991S}$. The treatment with 50 μM diamide caused 47.0 ± 1.9% ($n = 40$) depletion of ER Ca$^{2+}$ load for hRyR2$^{WT}$ and 28.9 ± 1.9% ($n = 48$) for hRyR2$^{C2991S}$ (Fig. 7d). Overall, the relative protective effect of the C2991S mutation was ~40% for both diamide concentrations (corresponding $p = 5.5 \times 10^{-7}$ and $p = 2.7 \times 10^{-9}$). The level of protection was similar to those seen for the hRyR2$^{C1078S}$ mutant, suggesting a common mechanism for the redox resistance in hRyR2$^{C1078S}$ and hRyR2$^{C2991S}$ mutants. Thus, cysteines 1078 and 2991 are likely involved in the hRyR2 intersubunit disulfide formation.

To directly test that these residues provide thiol groups to form an intersubunit disulfide bond, we created a double mutant with

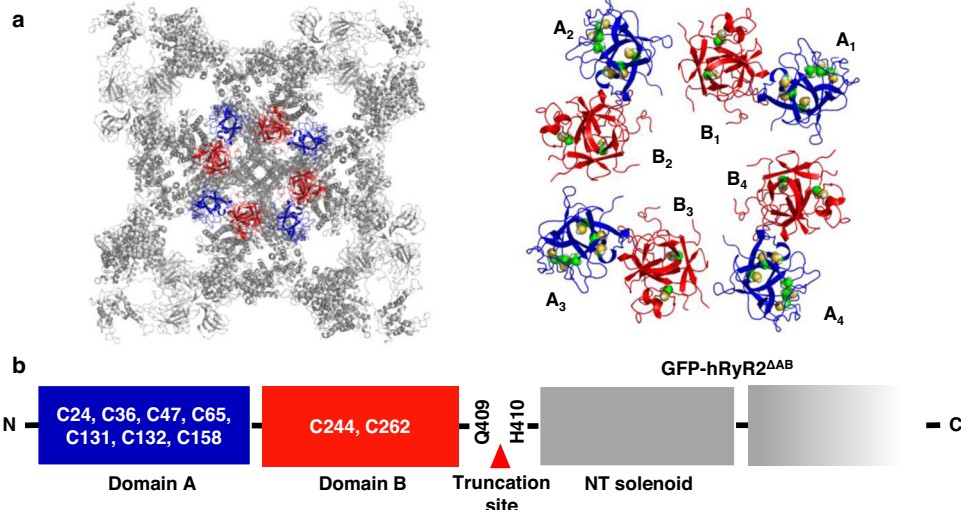

**Fig. 2 | Localization of the N-terminal domains A and B in RyR2. a** 3D visualization of the human RyR2 cryo-EM model (PDB ID: 7UA5) is shown in grey. Domains A and B are shown in blue and red, respectively. Cysteine residues are shown in green. **b** Localization of the truncation site in the AB-truncated hRyR2 mutant (hRyR2$^{\Delta AB}$) on the primary structure.

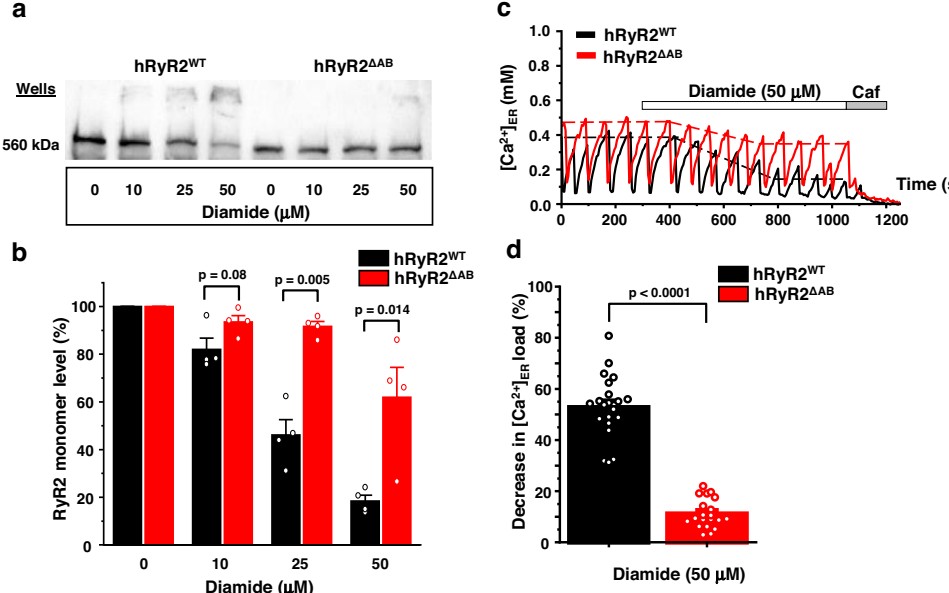

**Fig. 3 | Truncation of the N-terminal domains A and B in RyR2 inhibits intersubunit disulfide cross-linking and diamide-induced depletion in [Ca2+]ER.** **a** Representative western blot image of the hRyR2 cross-linking caused by diamide in HEK293 cells expressing GFP-hRyR2 and hRyR2$^{\Delta AB}$. All representative western blot images are presented as a top part, above the MW of 250 kDa, including the loading wells area. **b** Summary of the effect of the increasing diamide concentrations on the level of RyR2 monomeric form for hRyR2$^{WT}$ and hRyR2$^{\Delta AB}$ channels expressed in HEK293 cells. Data are shown as means ± SE ($n = 4$, each n represents data from an independent experiment) and were analyzed using a two-sided

unpaired t-test. **c** Representative recordings of Ca$^{2+}$ waves from HEK293 cells expressing hRyR2$^{WT}$ (black) and hRyR2$^{\Delta AB}$ (red) in control condition followed by the application of 50 μM diamide. The dashed lines indicate the level of ER Ca$^{2+}$ load. Caffeine (Caf; 10 mM) and ionomycin (2 μM) in 10 mM Ca$^{2+}$ were used to normalize the signal. **d** Summary of the effect of 50 μM diamide on ER Ca$^{2+}$ load in HEK293 cells transfected with hRyR2$^{WT}$ (black) or hRyR2$^{\Delta AB}$ (red). Data are shown as means ± SE ($n = 22$ for hRyR2$^{WT}$ and 20 cells for hRyR2$^{\Delta AB}$) and were analyzed using a two-sided unpaired t-test.

cysteines C1078 and C2991 mutated to serines (hRyR2$^{C1078S/C2991S}$). When expressed in HEK293 cells, the hRyR2$^{C1078S/C2991S}$ mutant demonstrated significant protection against hRyR2 intersubunit cross-link induced by diamide. The protection of hRyR2$^{C1078S/C2991S}$ monomers from cross-linking was in line with the one seen for hRyR2$^{C1078S}$ and hRyR2$^{C2991S}$ mutants: $53.7 \pm 7.6\%$ ($n = 5$, $p = 2.9 \times 10^{-5}$) at 25 μM diamide and $53.1 \pm 3.3\%$ ($n = 5$, $p = 2.4 \times 10^{-6}$) at 50 μM diamide (Fig. 8a, b). Because there was no further protection against hRyR2 cross-linking in

the double cysteine mutant, we concluded that cysteines 1078 and 2991 partner with one another to create a disulfide bond between neighboring hRyR2 subunits.

Next, we studied the effect of the simultaneous C1078S and C2991S mutations on the magnitude of [Ca$^{2+}$]$_{ER}$ depletion caused by diamide. Cells expressing the hRyR2$^{C1078S/C2991S}$ mutant generated Ca$^{2+}$ waves similar to those seen in cells expressing hRyR2$^{WT}$ (Fig. 8c and Supplementary Fig. 5). The treatment with 25 μM diamide caused the

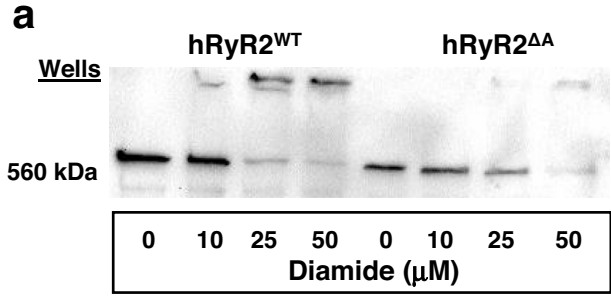

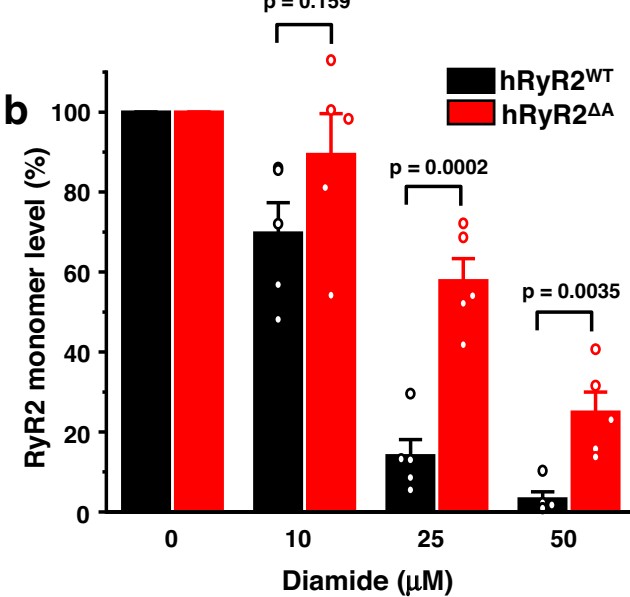

**Fig. 4 | N-terminal domain A of hRyR2 has an allosteric effect on cross-linking.**
**a** Representative western blot image of hRyR2 cross-linking caused by diamide in HEK293 cells expressing hRyR2$^{WT}$ and hRyR2$^{\Delta A}$. **b** Summary of the effect of the increasing diamide concentrations on the level of RyR2 monomeric form for hRyR2$^{WT}$ and hRyR2$^{\Delta A}$ channels expressed in HEK293 cells. Data are shown as means ± SE ($n = 5$, each n represents data from an independent experiment) and were analyzed using a two-sided unpaired t-test.

48.6 ± 2.0% ($n = 39$) decline in ER Ca$^{2+}$ load for hRyR2$^{WT}$ and 24.3 ± 1.6% ($n = 44$) for hRyR2$^{C1078S/C2991S}$ (Fig. 8d). Corresponding values for 50 μM diamide were 60.8 ± 2.1% ($n = 24$) for hRyR2$^{WT}$ and 28.3 ± 1.9% ($n = 27$) for hRyR2$^{C1078S/C2991S}$ (Fig. 8d). Overall, the simultaneous cysteine 1078 and 2991 mutations prevented the ER Ca$^{2+}$ load depletion by ~50% for both diamide concentrations ($p = 3.5 \times 10^{-15}$ and $p = 2.7 \times 10^{-15}$). This level of protection against the cross-linking was similar to those seen for the single cysteine mutants (Figs. 6, 7), confirming that cysteines 1078 and 2991 form the functionally important intersubunit disulfide bond within the hRyR2 complex.

This study reveals that out of 89 cysteines in hRyR2, two cysteines (C1078 and C2991) play a particularly important role in the functional response of the channel to oxidative stress. Mutation of one of those cysteines substantially protected against hRyR2 cross-linking and ER Ca$^{2+}$ leak activation caused by diamide. Notably, we observed a similar protective effect of hRyR2 C1078S and C2991S mutations in cells treated with increasing hydrogen peroxide (H$_2$O$_2$) concentrations (Supplementary Fig. 6). While diamide is an effective exogenous agent that specifically oxidizes thiol groups, H$_2$O$_2$ is an endogenous oxidant produced in vivo during oxidative stress[42,43]. Therefore, the redox response caused by cysteines 1078

and 2991 oxidation represents a universal reaction of RyR2 to oxidative stress.

## Discussion

In the mammalian heart, SR Ca$^{2+}$ released through RyR2 plays a central role in initiating a robust myocardial contraction. Consequently, defects in RyR2 function can lead to contractile dysfunction in a variety of cardiac pathologies. It has been shown that redox-dependent PTMs of RyR2 contribute to abnormal Ca$^{2+}$ regulation during myocardial infarction and heart failure[4–7]. In the canine model of chronic heart failure, disulfide formation within the RyR2 complex has been characterized as the major redox modification responsible for "leaky" RyR2, abnormal Ca$^{2+}$ regulation, and arrhythmias[5]. We previously showed that oxidative stress could promote the formation of disulfide bonds between two neighboring subunits of RyR2 through a PTM known as intersubunit cross-linking[22]. We found a strong positive correlation between the level of RyR2 cross-linking and an elevated SR Ca$^{2+}$ leak[7,13,22]. Moreover, a substantial portion of RyR2 in cardiomyocytes from failing hearts exists in the cross-linked form[7]. Despite its clinical significance, the redox-sensitive sites involved in RyR2 cross-linking have not yet been identified.

By analyzing the recently published cryo-EM structures of hRyR2[41], we identified two candidates for the channel's intersubunit disulfide formation at neighboring subunits of RyR2: cysteines 1078 and 2991 (Fig. 5 and Supplementary Fig 4a). We found that C2991 on the HD2 loop is solvent accessible in both closed and open conformations (Fig. 5 and Supplementary Fig 4b, Supplementary Table 1). Although a limited resolution in the area of the P1-SPRY2 linker does not provide structural information about the precise localization of C1078, this cysteine might be also accessible to oxidation (Fig. 5 and Supplementary Fig 4c, Supplementary Table 1). Therefore, based on structural localization and solvent accessibility, we decided to study the role of cysteines 1078 and 2991 in RyR2 disulfide cross-linking and functional response to oxidative stress. We discovered that mutations of either one of those cysteines or the mutation of both substantially protected against hRyR2 cross-linking caused by diamide or H$_2$O$_2$. To get further insights into the mechanism of the C1078-C2991 disulfide formation, we analyzed orientation and solvent accessibility of these cysteines at different RyR2 conformations, including phosphorylation by protein kinase A (PKA), CPVT mutation, and the presence of CaM or the Rycal drug ARM210[41] (for the detailed analysis see Supplementary Table 1 and Supplementary Figs. 4, 7–9). The analysis revealed that the C2991 thiol group is accessible to a solvent, with higher accessibility in open and phosphorylated conformations. The binding of CaM or ARM210 also increases the solvent accessibility. However, an introduction of the CPVT R2474S mutation decreases the accessibility, especially in the open conformation. Due to its limited resolution, we could not reliably evaluate the conformation of the P1-SPRY2 linker and C1078 orientation in different RyR2 models. However, this residue could be accessible for oxidation due to the high levels of the linker's flexibility. Even though available cryo-EM models[41] do not indicate the favorable mutual orientation of C1078 and C2991 for the disulfide bond formation, we believe this bond is sterically possible in some P1-SPRY2 linker conformations (Supplementary Fig. 10)[40,41]. We propose a model where an oxidant (diamide, oxidized glutathione) would bind to C1078 and stabilize the linker in the favorable conformation for further interaction with C2991. In the next step, C1078 would create a disulfide bond with C2991, releasing the reduced form of oxidant (hydrazine, glutathione). The disulfide bond formation between two RyR2 subunits highlights the essential functional role of the intersubunit interface comprised of the P1-SPRY2 linker from one subunit and the HD2 loop from another subunit. In the previous structural study of the RyR2$^{R2474S}$ CPVT mutant[41], ARM210 has been shown to bind to the P1 domain, stabilizing the

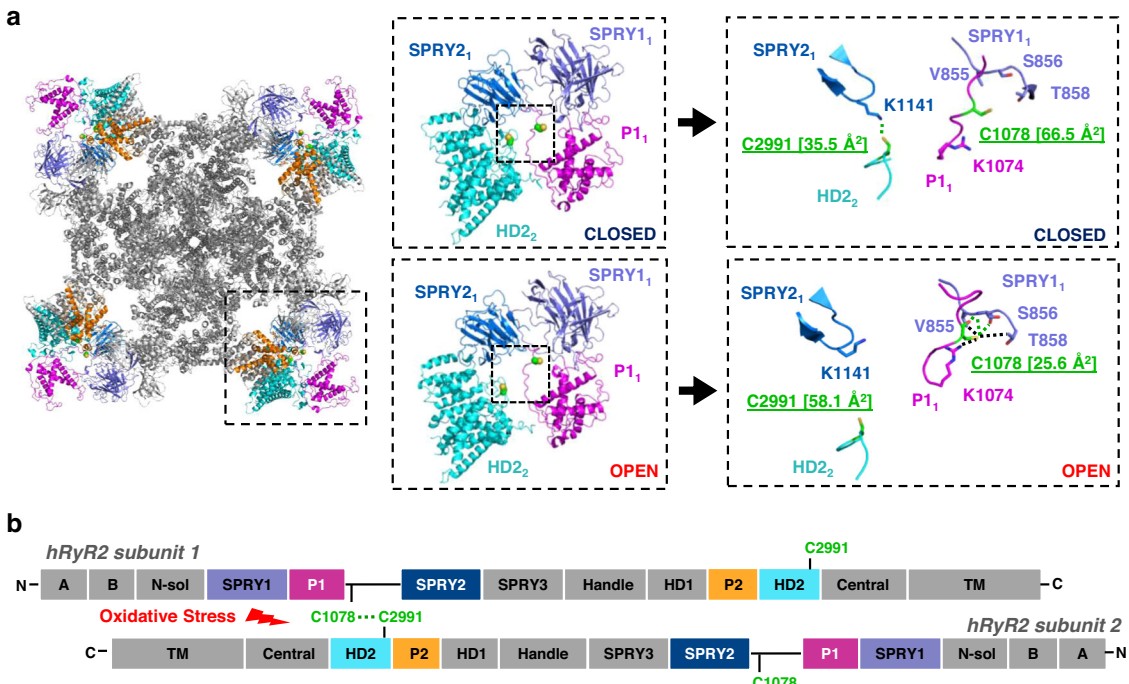

**Fig. 5 | Cysteines 1078 and 2991 face one another at the neighboring subunits of hRyR2 and are accessible for oxidation depending on channel conformation. a** Suggested localization of cysteines 1078 and 2991 in the 3D cryo-EM model of the closed hRyR2 (PDB ID: 7UA5). Helical domain 2 is shown in cyan. P1 and P2 domains are shown in magenta and orange, respectively. SPRY1 and SPRY2 domains are shown in slate and marine, respectively. The cross-linking cysteines 1078 and 2991 are shown in green. The right panel illustrates the local environments of C1078 and

C2991 in closed (PDB ID: 7UA5) and open (PDB ID: 7UA9) conformations. The number in the square brackets represents the cysteine thiol group's solvent-accessible surface area (ASA) in Å2. Black dashed lines between residue side chains represent the van der Waals interactions, and green dashed lines represent hydrogen bonds. **b** Localization of cysteines 1078 and 2991 on the primary hRyR2 structure and a potential site of the disulfide formation caused by oxidative stress.

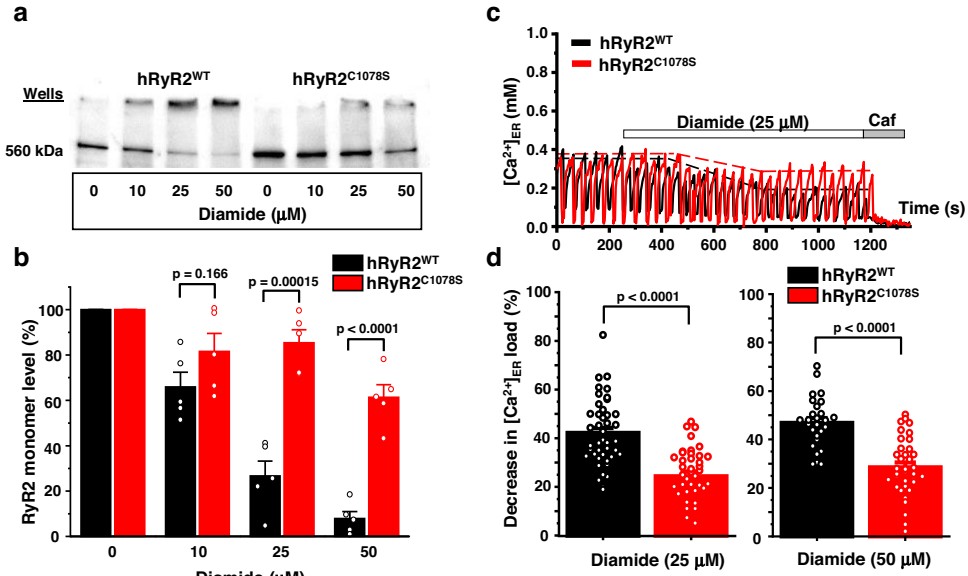

**Fig. 6 | Cysteine 1078 directly participates in hRyR2 cross-linking and functional activation by diamide. a** Representative western blot image of hRyR2 cross-linking caused by diamide in HEK293 cells expressing hRyR2$^{WT}$ and hRyR2$^{C1078S}$. **b** Summary of the effect of the increasing diamide concentrations on the level of RyR2 monomeric form for hRyR2$^{WT}$ and hRyR2$^{C1078S}$ channels expressed in HEK293 cells. Data are shown as means ± SE ($n = 5$, each n represents data from an independent experiment) and were analyzed using a two-sided unpaired t-test. **c** Representative recordings of Ca$^{2+}$ waves from Flp-In T-Rex-293 SERCA2a stable

line cells expressing hRyR2$^{WT}$ (black) and hRyR2$^{C1078S}$ (red) in control condition followed by the application of 25 μM diamide. Caffeine (Caf; 10 mM) and ionomycin (2 μM) in 10 mM Ca$^{2+}$ were used to normalize the signal. **d** Summary of the effect of 25 and 50 μM diamide on ER Ca$^{2+}$ load in cells transfected with hRyR2$^{WT}$ (black) or hRyR2$^{C1078S}$ (red). Data are shown as means ± SE (25 μM diamide: $n = 43$ for hRyR2$^{WT}$ and 40 cells for hRyR2$^{C1078S}$; 50 μM diamide: $n = 27$ for hRyR2$^{WT}$ and 36 cells for hRyR2$^{C1078S}$) and were analyzed using a two-sided unpaired t-test.

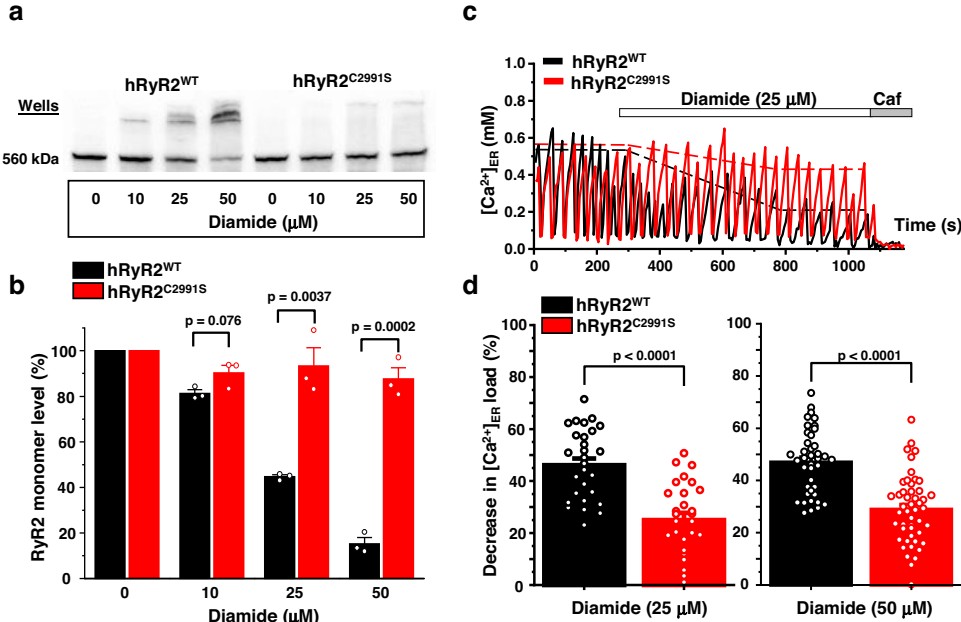

**Fig. 7 | Cysteine 2991 directly participates in hRyR2 cross-linking and functional activation by diamide. a** Representative western blot image of hRyR2 cross-linking caused by diamide in HEK293 cells expressing hRyR2^WT and hRyR2^C2991S. **b** Summary of the effect of the increasing diamide concentrations on the level of RyR2 monomeric form for hRyR2^WT and hRyR2^C2991S channels expressed in HEK293 cells. Data are shown as means ± SE ($n = 3$, each n represents data from an independent experiment) and were analyzed using a two-sided unpaired t-test. **c** Representative recordings of Ca²⁺ waves from Flp-In T-Rex-293 SERCA2a stable line cells expressing hRyR2^WT (black) and hRyR2^C2991S (red) in control condition followed by the application of 25 μM diamide. Caffeine (Caf; 10 mM) and ionomycin (2 μM) in 10 mM Ca²⁺ were used to normalize the signal. **d** Summary of the effect of 25 and 50 μM diamide on ER Ca²⁺ load in cells transfected with hRyR2^WT (black) or hRyR2^C2991S (red). Data are shown as means ± SE (25 μM diamide: $n = 29$ for hRyR2^WT and 28 cells for hRyR2^C2991S; 50 μM diamide: $n = 40$ for hRyR2^WT and 48 cells for hRyR2^C2991S) and were analyzed using a two-sided unpaired t-test.

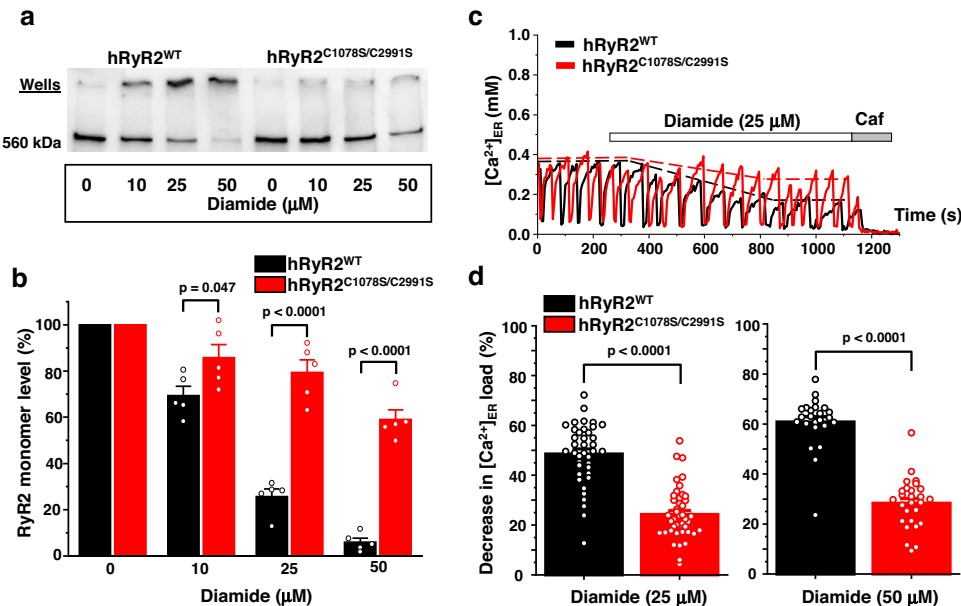

**Fig. 8 | Cysteines 1078 and 2991 promote hRyR2 activation during oxidative stress by forming a disulfide bond. a** Representative western blot image of hRyR2 cross-linking caused by diamide in HEK293 cells expressing hRyR2^WT and hRyR2^C1078S/C2991S. **b** Summary of the effect of the increasing diamide concentrations on the level of RyR2 monomeric form for hRyR2^WT and hRyR2^C2991S channels expressed in HEK293 cells. Data are shown as means ± SE ($n = 5$, each n represents data from an independent experiment) and were analyzed using a two-sample t-test. **c** Representative recordings of Ca²⁺ waves from Flp-In T-Rex-293 SERCA2a stable line cells expressing hRyR2^WT (black) and hRyR2^C1078S/C2991S (red) in control condition followed by the application of 25 μM diamide. Caffeine (Caf; 10 mM) and ionomycin (2 μM) in 10 mM Ca²⁺ were used to normalize the signal. **d** Summary of the effect of 25 and 50 μM diamide on ER Ca²⁺ load in cells transfected with hRyR2^WT (black) or hRyR2^C1078S/C2991S (red). Data are shown as means ± SE (25 μM diamide: $n = 39$ for hRyR2^WT and 44 cells for hRyR2^C1078S/C2991S; 50 μM diamide: $n = 24$ for hRyR2^WT and 27 cells for hRyR2^C1078S/C2991S) and were analyzed using a two-sided unpaired t-test.

intersubunit interface between the beginning of the P1-SPRY2 linker (preceding C1078) from one subunit and the HD2 loop (including C2991) from another subunit, reverting the primed state of the mutant channel to the closed state. In the proposed model, the stabilization of the intersubunit interface occurs through the interaction between D1070 on one RyR2 subunit and H2995 on another, with the possible involvement of H1071 and R2988[41]. The intersubunit disulfide formation between C1078 and C2991 would likely destabilize this interface, shifting the channel's structure to a "leaky" conformation.

By analyzing $[Ca^{2+}]_{ER}$ dynamics in HEK293 cells expressing hRyR2 and SERCA2a $Ca^{2+}$ pump, we showed that the mutations of cross-linking cysteines (C1078S or/and C2991S) substantially (~ 50 %) protect against a depletion of $[Ca^{2+}]_{ER}$ induced by oxidative stress. Since the level of oxidative stress that induced almost complete hRyR2[WT] cross-linking (at 50 μM diamide) did not affect the function of SERCA2a[29] and endogenous ion channels/pumps in our cell model system, the depletion of ER $Ca^{2+}$ load was chiefly mediated by increased ER $Ca^{2+}$ leak via hRyR2. It is well known that oxidation of RyR2 is commonly associated with increased channel activity and ER/SR $Ca^{2+}$ leak (reviewed in refs. 44,45). In the present study, we discovered that oxidation of only two cross-linking cysteines was responsible for half of the channel's functional response to oxidative stress. It appears that defective intersubunit dynamics due to RyR2 intersubunit cross-linking plays a key role in RyR2 dysfunction during oxidative stress.

Discovered redox-sensitive cysteines 1078 and 2991 are conserved in RyR2 but are missing in RyR1, indicating a unique redox-sensitive pair for the cardiac isoform of the channel (Supplementary Fig. 11). They contribute to the ~50% of RyR2 functional activation during oxidative stress, which raises a question about the additional RyR2 sites involved in redox signaling of the channel. It has been shown that the RyR2 molecule comprises more than 30 cysteines accessible to redox modifications[46,47]. In addition to disulfide formation, the mechanisms of direct RyR2 regulation by oxidative stress may include a sulfonic acid formation, *S*-nitrosylation and *S*-glutathionylation[46,48,49]. Alternatively, the indirect mechanism of redox regulation might can be mediated by CaM dissociation from the RyR2 complex[11–13]. Recently, it has been shown that RyR2 redox regulation also occurs in the SR lumen of cardiomyocytes, where oxidative stress disrupts the redox-dependent association between RyR2 and ERp44, through the upregulation of Ero1α reductase[50].

Although the NTR of hRyR2 was shown to be essential for the structural integrity of the channel[33,38], our experiments revealed that the individual cysteines from this region were not directly involved in RyR2 intersubunit cross-linking. At the same time, we believe that NTR domains A and B of RyR2 have an allosteric effect on hRyR2 structural re-arrangements during oxidative stress. The complete removal of domains A and B substantially reduced the levels of hRyR2 cross-linking and prevented ER $Ca^{2+}$ leak caused by oxidative stress. It appears the intact structure of the NTR is required for two cross-linking cysteines in neighboring RyR2 subunits to be in close proximity for a formation of disulfide bond. It has been shown that recombinant RyR2 can self-associate in an organized manner[51], suggesting a possibility for channel-channel cross-linking. However, both the P1-SPRY2 linker harboring C1078 and the loop that includes C2991 are parts of the RyR2 inner intersubunit area and do not belong to the proposed channel-channel self-organization interfaces[41].

In conclusion, the results of this study revealed two critical redox-sensitive residues in human RyR2. We found that, out of 89 cysteines in each hRyR2 subunit, cysteines 1078 and 2991 play a critical role in the channel's functional response to oxidative stress. Given the structural orientation on the neighboring RyR2 subunits, we suggest that cysteines 1078 and 2991 form a redox sensor pair capable of forming an intersubunit disulfide bond during oxidative stress, resulting in

structural remodeling of RyR2 and the formation of pathological "leaky" channel.

## Methods

### Vector production

The production of vectors encoding the RyR2, R-CEPIA1er, and SERCA2a has been previously described in ref. 29. The vector encoding human RyR2 cDNA fused to a green fluorescent protein (GFP) at the N-terminus was a gift from Dr. Christopher George (University of Cardiff). pCMV R-CEPIA1er was kindly provided by Dr. Masamitsu Iino (Addgene plasmid 58216; http://n2t.net/addgene: 58216; RRID: Addgene_58216). The vector encoding human SERCA2a cDNA was a gift from Dr. David Thomas (University of Minnesota). The SERCA2a cDNA was cloned into the mCerulean-M1 modified plasmid (Addgene, plasmid # 15214), yielding SERCA2a recombinant protein fused to a modified cerulean fluorescent protein (mCer) at the $NH_2$-terminus. The mutations in hRyR2 (hRyR2[ΔAB], hRyR2[ΔA] hRyR2[C24S], hRyR2[C36S], hRyR2[C47S/C65S], hRyR2[C131S/C132S/C158S], hRyR2[C1078S], hRyR2[C2991S], hRyR2[C1078S/C2991S]) were introduced by PCR using specific primers containing the mutations (IDT, USA. Sequences of these primers are listed in the Supplementary Table 2), and the Q5 mutagenesis kit (Nee England Biolabs, USA). After verification of the mutagenesis by single-pass analysis, the plasmids were amplified and used for experimentation. To generate the C131S/C132S *RyR2* mutant we used primers simultaneously containing both point mutations. However, all the other multiple cysteine *RyR2* mutants were generated by multi mutagenesis steps. Every new mutation was introduced using *RyR2* already containing mutations at other residues as a backbone.

### Measurements of RyR2 cross-linking

HEK293 cells (AAV pro 293 T Cell Line, Takara, #632273) were grown on 100 mm plastic dishes for 24 h and then transfected with the wild-type or mutant GFP-hRyR2 plasmid using polyethylenimine (PEI, 1 μM/ml). Cells were further grown for another 48 h. Cells were harvested and resuspended in the solution containing (in mM): 150 K-aspartate, 0.25 $MgCl_2$, 0.1 EGTA, 10 HEPES, and pH 7.2. The cells were treated with corresponding diamide concentrations, harvested, and resuspended in lysis buffer (150 mM NaCl, 25 mM Hepes pH 7.4, 1% Triton X-100, phosphatases inhibitors) with 5 mM N-ethylmaleimide (NEM) to block free sulfhydryl groups. Lysate samples were incubated with non-reducing Laemmli buffer for 10 min, ran on 2–10% gradient SDS-PAGE, and blotted overnight onto the nitrocellulose membrane. Cross-linking was detected using F1 anti-RyR2 primary antibody (1:1000; sc-376507, Santa Cruz, U.S.A.) and HRP-conjugated goat anti-mouse secondary antibody (1:5000; 1430, Thermo Fisher Scientific, U.S.A.). For N-truncated hRyR2 mutants rabbit polyclonal anti-GFP antibody (1:1000; A6455, Thermo Fisher Scientific, U.S.A.) was used as primary, and HRP-conjugated goat anti-rabbit antibody (1:5000, A0545, Sigma-Aldrich Co, U.S.A.) as secondary. The amount of hRyR2 cross-linking was analyzed in ImageJ software (NIH, USA) based on the disappearance of the monomeric 560 kDa band using the densitometry analysis.

### Generation of the inducible T-Rex SERCA2a stable cell line

The generation of stable inducible Flp-In T-Rex-293 cell line expressing SERCA2a using the Flp-In T-REx Core Kit (Invitrogen) has been previously described in ref. 29. Briefly, *SERCA2a* cDNA was cloned into the inducible expression vector pcDNA5/FRT/TO. Obtained vector, together with the pOG44 vector encoding the Flp recombinase, was used to co-transfect Flp-In-T-Rex293 cells (R780-07, Thermo Fisher Scientific, U.S.A.). Forty-eight hours after transfection, the growth medium was replaced with a selection medium containing 100 μg/ml hygromycin (Invitrogen). The selection medium was changed every 2 days until the stable line cell foci could be isolated.

The hygromycin-resistant cell foci were selected and expanded. Stable line cells were cultured in high-glucose DMEM supplemented with 100 U/ml penicillin, 100 mg/ml streptomycin, and 10% fetal bovine serum at 5% $CO_2$ and 37 °C.

## Measurements of the ER luminal [Ca²⁺] ([Ca²⁺]$_{ER}$)

For functional $Ca^{2+}$ measurements we used HEK293 or T-Rex SER-CA2a stable cells. Stable cells were co-transfected with plasmids containing *GFP-hRyr2* and *R-CEPIA1er* genes. HEK293 cells were also transfected with the plasmid containing the *mCer-SERCA2a* gene to ensure the physiological ER $Ca^{2+}$ concentrations. Experiments were carried out 48 h after transfection when expression of the exogenous genes was optimal. All experiments were conducted using laser scanning confocal microscopy (Radiance 2000 MP, BioRad, UK) equipped with a 40x oil objective lens (N.A. = 1.3). The cells prepared for an experiment were bathed in Tyrode solution containing (in mM): 135 NaCl, 4 KCl, 2 $CaCl_2$, 1 $MgCl_2$, 10 D-glucose, and 10 HEPES. To record changes in [Ca²⁺]$_{ER}$, R-CEPIA1er was excited with the 543 nm line of a He-Ne laser, and fluorescence was measured at > 580 nm. For measurements of [Ca²⁺]$_{ER}$ load-leak balance, 2D images (512 × 512 pixels) were collected every 5 s at a scanning speed of 6 ms/line. The cells were treated with corresponding diamide concentration for at least 15 min until the complete equilibration of [Ca²⁺]$_{ER}$ levels. At the end of each experiment, caffeine (10 mM) was applied to induce complete depletion of [Ca²⁺]$_{ER}$. Then, $F_{max}$ was estimated by applying 2 μM ionomycin, escin 0.005%, and 10 mM $Ca^{2+}$. The changes in [Ca²⁺]$_{ER}$ were presented normalized according to [Ca²⁺]$_{ER}$ = (F-$F_{min}$)/($F_{max}$-$F_{min}$). To obtain the change in [Ca²⁺]$_{ER}$ caused by oxidative stress, $Ca^{2+}$ wave peak values were averaged before and after diamide application, and the relative change was calculated.

## Statistics

All data are presented as means ± SEM of n measurements. Obtained 2D images were analyzed using ImageJ software (NIH, USA). Groups were compared using the two-sided Student's t-test for unpaired data sets or one-way ANOVA followed by Tukey's post-hoc test where appropriate. Differences were considered statistically significant at $P < 0.05$. Peak analysis, statistical analysis, and graphical representation of averaged data were carried out on Origin 2021b SR2 software (OriginLab, U.S.A.).

## Structural analysis and model visualization

The cryo-EM models of RyR2 were obtained from RSCB protein data bank. The structural analysis was performed using PyMOL 1.7.0.3, WinCoot 0.9.8.1 and AREAIMOL program from CCP4i package (v. 8.0.007). PyMOL 1.7.0.3 was used to create graphical representations of the RyR2 structural models.

## Reporting summary

Further information on research design is available in the Nature Portfolio Reporting Summary linked to this article.

# Data availability

The data supporting this article and other findings are available within the manuscript, figures, supplementary data, and from the corresponding authors upon request. The cryo-EM models of RyR2 used in this study 7UA5, 7UA9, 7U9T, 7U9Q, 7U9R, 7UA3, 7UA1, 7U9X, 7UA4, 7U9Z, 6WOU, 6WOV were obtained from Protein Data Bank. Source data are provided with this paper.

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

## Acknowledgements

This work was supported by the National Institutes of Health Grants R01HL151990 (to A.V.Z.). The authors would like to thank Dr. Christopher George (University of Cardiff, UK) for providing the vector encoding the human RyR2. The authors also would like to thank Dr. Masamitsu Iino for donating the R-CEPIA1er vector.

## Author contributions

R.N. and A.V.Z. conceived and supervised the study. R.N., E.B., R.G., T.J., and A.V.Z designed and performed Ca2+ imaging and biochemistry experiments. E.B. and D.K. performed DNA cloning experiments. R.N., E.B., R.G., and T.J. performed data analysis. R.N. and A.V.Z. wrote the manuscript. All the authors read and approved the manuscript version to be published.

## Competing interests

The authors declare on competing interests.
