## [Peer Review File · Nature Communications]

Cysteines 1078 and 2991 cross-linking plays a critical role in redox regulation of cardiac RyRReviewers' Comments:

Reviewer #1:

Remarks to the Author:

Overall, Zima and coworkers do a good job at identifying two Cys (out of 89) present in RYR2 that seem to contribute to diamine mediated intersubunit cross linking via disulfide bond formation.

Although there are many strengths to the work presented in the manuscript and the methodological approaches are innovative, several major weaknesses are noted that must be addressed before further consideration.

1. The concentration of diamine used are saturating and are likely to crosslink several if not all the free Cys in the preparation. the stoichiometry between moles of diamine to moles of RYR2 free Cys must be established, especially since the authors make an implicit assumption that they have selectively cross linked two pairs of RYR2 Cys. I don't think this assumption can be conclusive without chemical evidence via mass spectrometry. Why would 50uM diamine be expected to be selective toward cross linking two out of thousands of free Cys in the cellular preparation being studied?

2. The authors use caffeine to test the state of the stores at the end of each experiment. I believe that this protocol is intended to test the filling state of RYR2-sensitive ER store. However, with our parallel experiments testing the ER store filling state with TG or CPA it is not possible to definitely conclude whether the ER stores are depleted or whether the RYR2 constructs expressed are simply less able to respond to caffeine challenge.

3. Although the models used are clever, their translational relevance to human cardiomyopathies is conjectural at best.

Minor

the authors completely ignore the seminal work of Abramson, Salama, Pessah ect...on hyperactive RYR Cys.

Reviewer #2:

Remarks to the Author:

In this manuscript, Nikolaienko et al investigate the oxidation of the cardiac Ryanodine Receptor (RyR2). Faulty regulation of RYR2 has long known to have pathological consequences. One long-standing issue has been the effect of oxidation on RyR2 function, as RyR2 oxidation, as a result of oxidative stress, is thought to increase RyR2 opening, resulting in pathological 'leak' of calcium from the SR into the cytosol. Previous work from the authors has suggested that RyR2 can form intersubunit disulfide bonds under oxidizing conditions, and in this study they attempted to pin down the exact cysteine residues that are responsible. They conclude that a disulfide bond between two specific cysteines (C1078 and C2991) is responsible for 50% of the increased calcium leak. Using HEK293 cells expressing RyR2 and SERCA2a as a model system, the authors employed a combination of gel-based assays to determine whether RyR2 is in a monomeric versus multimeric (cross-linked state) and calcium imaging to monitor ER calcium content.

Overall, the precise mechanism of oxidation-dependent modulation of RyR2 is an important puzzle to solve. Identifying an intersubunit disulfide bond that can result in increased RyR2 opening is thus an important step forward. My enthusiasm is a bit dampened, however, as there is not much more information than specifying two cysteines that are involved (and excluding a couple of others), and there is a lack of mechanistic description that shows how this particular disulfide bond can result in increased channel opening. The manuscript could benefit from a more detailed description of the

cysteines and their environment, and should also discuss the possibility for alternative models that may explain the data.

1) The following is critical for proper assessment of the results: A great deal of this work relies on western blots, showing 'disappearance' of the monomeric RyR2 band. Although the figures do show a bit of the region just above, with appearance of a higher MW band, the reviewer cannot assess whether there are other, higher MW bands beyond what is shown. It is critical that the authors show the entire western blots (e.g. as supplementary), as it is possible that there are even higher-MW aggregates that would lead to a completely different interpretation. Are there signs of some RyR2 getting stuck in the loading wells? This should be shown explicitly. During SDS-PAGE, further cross-links can happen between adding SDS and running the gel. Although the authors used NEM to block free cysteines, this only applies to cysteines that are solvent accessible in the folded form, not buried cysteines that only become exposed upon denaturation with SDS. Thus, full western blots are needed before this reviewer can truly assess whether the interpretation (a specific intersubunit, but intra-RyR2, disulfide bond) is appropriate.

2) The authors show that the two proposed cysteines are in physical proximity. However, proximity is not the only criterion to allow for disulfide bond formation, and parameters such as surface accessibility of the SH group should be analyzed. I also urge the authors not to use a low-resolution RyR2 structure as a reference (the one used has a reported resolution of 5.1Å, at which level side chains are not visible). Instead, the authors could rely on a number of structures recently published by Miotto et al (2022; *Sci. Advances*), which are at resolutions better than 3Å. The accessible surface area of the Cysteine side chains should be reported. This should be done for RyR2 in multiple conformational states, as the accessibility may change according to the overall conformation. A figure zooming in on Cys2991, showing its packing and precise chemical environment should be shown instead of the zoomed-out versions currently shown. As currently shown, these do not provide sufficient detail to show whether the cysteines are truly accessible for disulfide bond formation.

3) RyRs are known to form higher-order interactions with neighboring RyRs, often in a regular, 2D-crystalline manner. The Cys2991 is close to a proposed inter-RyR interface. How are the authors certain that they are not looking at inter-RyR interactions? Or a combination of intra-RyR (but intersubunit) and inter-RyR disulfide bonds? The SDS-PAGE results cannot discriminate between these possibilities. Unless there is other evidence, this possibility should be discussed.

4) The mechanistic implication of an inter-subunit disulfide bond is only vaguely suggested. The authors did not capitalize on various available cryo-EM structures of both RyR1 and RyR2 in different states. As C1078 is in a flexible linker, invisible in most RyR cryo-EM structures, there is considerable 'slack' to allow this Cys to approach e.g. Cys2991. A straightforward mechanism would be that the distance between the two cysteines is too large to allow disulfide bond formation in a particular conformation of the RyR. I strongly urge the authors to take high-resolution structures of RyR2 in distinct conformations, and model in the flexible loop with Cys2991, seeing if it can physically reach Cys1078, while satisfying all geometrical parameters (bond lengths and angles, VDW clashes etc). If this is sterically possible in some, but not all states, then this immediately provides a mechanism. Another possibility is that the surface accessibility of the cysteines changes as a function of conformation, which also would reveal a mechanism.

As oxidation of RyRs goes hand-in-hand with disease-associated mutations, the authors could also take a look at disease mutant structures, as the Helical domain has been shown to undergo structural changes as a result of mutation in both RyR1 and RyR2. Finally, CaM binds to the helical domain, close to Cys2991, and cryo-EM studies on RyR1 and RyR2 bound to CaM have also shown conformational

changes of the helical domain containing Cys2991. Especially since CaM is natively expressed in HEK293 cells, disulfide bond formation may thus affect CaM binding, therefore not allowing CaM modulation. Thus, the authors should also assess the sterical constraints to form disulfide bonds in CaM-bound conformations.

5) The authors started out with removing the N-terminal domains (A and B) of the RyR2. This seemed to have a big effect on the ability of RyR to form intersubunit disulfide bonds, but individual mutations in these domains failed to reproduce this effect. The authors concluded that removal of these domains resulted in allosteric effects. Removal of entire domains is indeed very drastic and it is even surprising that it leads to functional channels. In fact, a previous study deleting these domains has already shown that this results in different functional behavior (Liu et al , 2015, J.Biol.Chem. 290, 7736-7743). Although already referenced, the authors can also particularly use this reference to strengthen the argument that deleting these domains causes allosteric changes.

In regards to the individual cysteines, the authors should have analyzed the precise environment of these, as they would have seen that they are not available for forming intersubunit interactions. Although they are somewhat 'close' on the scale of the full RyR2, they are in fact quite far from any Cys in a neighboring subunit, and there is quite some folded protein in between them. The result where the point mutations in these N-terminal domains have no effect on intersubunit disulfide bond formation is thus trivial and expected. The authors could still keep the corresponding data, but rather as a negative control, establishing the validity of the experimental approach.

Nature Communications manuscript NCOMMS-22-45497: "Cysteines 1078 and 2991 cross-linking plays a critical role in redox regulation of cardiac RyR".

We are thankful for the thoughtful suggestions and constructive criticisms provided by the reviewers. We have revised the manuscript to address all reviewer suggestions. A point-by-point response to each comment is provided below.

Reviewer #1:

1. *The concentration of diamine used are saturating and are likely to cross-link several if not all the free Cys in the preparation. The stoichiometry between moles of diamine to moles of RYR2 free Cys must be established, especially since the authors make an implicit assumption that they have selectively cross-linked two pairs of RYR2 Cys. I don't think this assumption can be conclusive without chemical evidence via mass spectrometry. Why would 50uM diamine be expected to be selective toward cross-linking two out of thousands of free Cys in the cellular preparation being studied?*

We have previously shown that the diamide concentration (50 μ M) that causes the maximal RyR2 intersubunit cross-linking produces only partial oxidation of the redox sensor roGFP¹, suggesting that RyR2 cross-linking cysteines are highly redox-sensitive residues in our cell model system. Second, the focus of this study was to identify RyR2 cysteines involved in cross-linking and to evaluate the functional significance of this post-translational modification. We found that the diamide concentration that causes the maximal RyR2 cross-linking did not affect ER Ca²⁺ leak/load in cells that do not express RyR2 (Supplementary information, Fig 2), suggesting this level of oxidative stress does not affect other ion channels/pumps in our cell system. If diamide did cause oxidation and cross-linking of other free cysteines outside RyR2, the importance of such redox modification in the regulation of ER Ca²⁺ would be negligible. Third, we studied the effect of diamide on ER Ca²⁺ leak in HEK293 cells acutely transfected with RyR2 DNA. Therefore, the concentration of RyR2 (i.e., expression level) varied significantly from cell to cell. To correct for this, we analyzed the effect of diamide/peroxide in individual cells as a relative decrease in ER Ca²⁺ load due to the activation of RyR2-mediated Ca²⁺ leak. Finally, it has been suggested that the oxidation of different classes of redox-sensitive cysteines in RyR2 would lead to different functional effects²⁻⁵. Our study focused on the single pair of cysteines (C1078 and C2991) critical for the RyR2 intersubunit cross-linking and ER Ca²⁺ leak. By introducing point mutations, we could assess these cysteines' structural and functional role independently from the oxidation of other thiol groups within RyR2. The protective effect of C1078 and C2991 mutations in cross-linking experiments indicates that oxidation of other free cysteines in RyR2 is unlikely to be involved in specific RyR2 intersubunit cross-linking.

2. *The authors use caffeine to test the state of the stores at the end of each experiment. I believe that this protocol is intended to test the filling state of RYR2-sensitive ER store. However, with our parallel experiments testing the ER store filling state with TG or CPA it is not possible to definitely conclude whether the ER stores are depleted or whether the RYR2 constructs expressed are simply less able to respond to caffeine challenge.*

As suggested by the reviewer, we performed additional experiments to verify whether caffeine (10 mM) causes complete depletion of ER Ca²⁺ load in our cell system. When RyR2 was activated by caffeine, SERCA inhibition with a high dose of thapsigargin (10 μ M) did not cause further depletion of ER Ca²⁺ load (Supplementary information, Fig 1). The same effects of caffeine and thapsigargin were observed for the hRyR2^{C1078S/C2991S} mutant. Thus, in our cell model, the application of 10 mM caffeine causes complete depletion of ER Ca²⁺ load and can be used to subtract the Ca²⁺-independent fluorescence of R-CEPIA1er. Moreover, a mechanism by which caffeine activates RyR2 is by increasing the channel sensitivity to

cytosolic Ca^{2+} ⁶. The analysis of Ca^{2+} -induced Ca^{2+} release (CICR) under control conditions revealed that RyR2^{WT} and different cysteine mutants had a comparable CICR frequency (Supplementary information, Fig 5), suggesting a similar RyR2 activity. We have added this information to the revised manuscript.

3. *Although the models used are clever, their translational relevance to human cardiomyopathies is conjectural at best.*

Oxidative stress plays a critical role in defective Ca^{2+} handling during cardiovascular diseases, including heart failure⁷. It has also been shown that post-translational redox modifications of RyR2 are associated with increased RyR2 activity⁸⁻¹². In this study, we used *in vitro* approaches to identify cysteines involved in the pathologically relevant redox modification of human RyR2, as significant level of RyR2 cross-linking has been detected in ventricular myocytes from failing hearts⁹. Thus, we believe that identifying the functionally important redox-sensing sites of RyR2 can help in the design of therapeutic interventions that can selectively protect Ca^{2+} regulation during oxidative stress.

Minor. *The authors completely ignore the seminal work of Abramson, Salama, Pessah ect...on hyperactive RyR Cys.*

In the revised manuscript, we added a brief discussion regarding the discovery of hyperactive cysteines by Drs. Abramson, Salama, and Pessah¹³.

Reviewer #2:

1a. *The following is critical for proper assessment of the results: A great deal of this work relies on western blots, showing 'disappearance' of the monomeric RyR2 band. Although the figures do show a bit of the region just above, with appearance of a higher MW band, the reviewer cannot assess whether there are other, higher MW bands beyond what is shown. It is critical that the authors show the entire western blots (e.g. as supplementary), as it is possible that there are even higher-MW aggregates that would lead to a completely different interpretation. Are there signs of some RyR2 getting stuck in the loading wells? This should be shown explicitly... Thus, full western blots are needed before this reviewer can truly assess whether the interpretation (a specific intersubunit, but intra-RyR2, disulfide bond) is appropriate.*

In our original submission, we provided the western blots' uncut top part (>250 kDa), including the loading wells of the protein gels. In most cases, the band representing the cross-linked form of RyR2 was stuck in the well. Sometimes the oligomeric forms of RyR2, due to the cross-linking, could travel a bit below the loading zone due to some variability in gel properties. We specified the loading wells area in our updated

Figure 1. Representative Western blots of RyR2^{WT} and cysteine mutants treated with increasing diamide concentrations.

figures to clarify this issue. Examples of the entire western blots for RyR2^{WT} and different cysteine mutants are shown here in Fig 1.

1b. *During SDS-PAGE, further cross-links can happen between adding SDS and running the gel. Although the authors used NEM to block free cysteines, this only applies to cysteines that are solvent accessible in the folded form, not buried cysteines that only become exposed upon denaturation with SDS.*

The focus of this study is to determine the functional significance of intersubunit cross-linking between cysteines 1078 and 2991. Although nonspecific cysteine interactions are still possible, we believe they have no significant effect on the data interpretation for several reasons:

1. During sample collection, we removed the solution with diamide or peroxide and replaced it with the lysis buffer containing 5 mM NEM. In the case of spontaneous disulfide formation upon the SDS treatment under ambient conditions, we would observe cross-linking in samples not treated with oxidants. However, we have never observed RyR2 cross-linking in control untreated samples.
2. We showed that the mutation of either cysteine residue (C1078 or C2991) could independently prevent oxidative cross-linking of RyR2. When new thiol groups become available for oxidation after RyR2 denaturation with SDS, removing a single cysteine would not be sufficient to prevent nonspecific cross-linking because of the second reactive cysteine.
3. Finally, the structural analysis of Cryo-EM models has shown that at least one of the cysteine residues (C2991) is mostly solvent accessible (please see the following comments #2) and would likely be oxidized or react with NEM before the SDS treatment.

Therefore, we believe that the discovered C1078 and C2991 are part of the specific RyR2 intersubunit cross-linking mechanism that is not affected by the spontaneous oxidation of other cysteines.

2. *The authors show that the two proposed cysteines are in physical proximity. However, proximity is not the only criterion to allow for disulfide bond formation, and parameters such as surface accessibility of the SH group should be analyzed. I also urge the authors not to use a low-resolution RyR2 structure as a reference (the one used has a reported resolution of 5.1 Å, at which level side chains are not visible). Instead, the authors could rely on a number of structures recently published by Miotto et al (2022; Sci. Advances), which are at resolutions better than 3 Å. The accessible surface area of the Cysteine side chains should be reported. This should be done for RyR2 in multiple conformational states, as the accessibility may change according to the overall conformation. A figure zooming in on Cys2991, showing its packing and precise chemical environment should be shown instead of the zoomed-out versions currently shown. As currently shown, these do not provide sufficient detail to show whether the cysteines are truly accessible for disulfide bond formation.*

In accordance with the reviewer's suggestions, we took advantage of the higher-resolution Cryo-EM models of hRyR2 published by the Marks group¹⁴ to analyze the surface accessibility of the cross-linking cysteines. These models provided valuable structural information about the cross-linking cysteine 2991. Despite the models' overall high resolution, the linker's resolution bearing another cysteine 1078 was not high enough to calculate its geometrical parameters in this area reliably (EM density was not visible at recommended settings). Thus, most of the following structural analysis is focused on C2991, with only limited data on C1078.

In our analysis, we assessed the solvent-accessible surface areas (ASA) for the thiol groups of these cysteines (Supplementary information, Table 1). For reference, the estimated maximal ASA for the cysteine's thiol group in the Glycine-Cysteine-Glycine tripeptide is 82.07 Å²¹⁵. Based on this value, we calculated the relative solvent accessibility (RSA) to estimate the relative exposure of SH-groups. We also

calculated ASA and RSA values for entire cysteine residues (Supplementary information, Table 1). Residues with an RSA value of more than 20% are considered exposed to solvent¹⁶.

The effect of hRyR2 gating states on the accessibility of cross-linking cysteines.

We compared the structures of closed and open dephosphorylated RyR2 with no bound calmodulin (CaM; PDB IDs: 7UA5; 7UA9). On these structures, C2991 is located at the loop of the HD2 domain close to the P1 – SPRY2 linker from another RyR2 subunit that includes C1078 (Supplementary information, Fig 4a). In the closed conformation, the SH-group of C2991 is oriented towards the SPRY2 domain in a way that makes a hydrogen bond with K1141 (Supplementary information, Fig 4a and b). This hydrogen bond contributes to the somewhat decreased but relatively high ASA of 35.5 Å². This value substantially improves up to 58.1 Å² during the RyR2 transition to the open conformation (Supplementary information, Fig 4b). This effect could result from the change in intersubunit distances between the SPRY2 domain on the first subunit and the HD2 domain on the second. Consequently, the distance between C2991 and K1141 increases, eliminating the ability to form a hydrogen bond.

We also analyzed the accessibility of cross-linking C1078. We found that its thiol group has a high ASA of 66.5 Å² in the closed RyR2 conformation and a decreased ASA of 25.6 Å² in the open conformation (Supplementary information, Fig 4c). The drop in ASA is mainly associated with the C1078 interactions with residues of the SPRY1-P1 linker (V855, S856, and T858) and neighboring R1074 on the same subunit. Unfortunately, due to limited resolution in this area, we cannot reliably establish a correlation between the RyR2 conformations and the C1078 accessibility to oxidation. However, we may conclude that the P1-SPRY2 linker has a high level of flexibility that may lead to C1078 thiol group exposure, at least in some conformations. The ASA and RSA values for C1078 residue are provided in Supplementary information Table 1.

The effect of PKA phosphorylation and CaM binding to hRyR2^{WT} on C2991 SH-group availability.

The comparison of dephosphorylated and phosphorylated models of RyR2^{WT} (PDB IDs: 7UA5; 7UA9; 7U9Q; 7U9R) showed that PKA phosphorylation increased C2991 thiol group ASA values in both conformations (Supplementary information, Fig 7a) from 35.5 to 45.9 Å² in the closed and from 58.1 to 63.3 Å² in the open conformation (Supplementary information, Fig 7b). Thus, PKA phosphorylation of hRyR2 would promote solvent exposure of the thiol group of C2991 in both conformations. However, the open conformation of hRyR2 has a higher potential for C2991 oxidation due to higher ASA values and a larger gap between subunits that is more accessible for the bulkier molecules of an oxidant, such as diamide. For phosphorylated hRyR2 in the closed conformation, the binding of CaM (PDB ID: 7U9T) caused a noticeable increase in the ASA value of the C2991 thiol group from 45.9 to 58.9 Å² (Supplementary information, Fig 7c).

The effect of R2474S CPVT mutation of hRyR2 on C2991 SH-group availability.

For the available Cryo-EM models of phosphorylated RyR2 (PDB IDs: 7U9Q; 7U9R), we found that an introduction of the point R2474S CPVT mutation (PDB IDs: 7U9X; 7U9Z) did not substantially change the C2991 thiol group ASA value in the closed conformation (Supplementary information, Fig 8a) but substantially decreased it in the open conformation from 63.3 to 29.4 Å² (Supplementary information, Fig 8b). This effect was achieved by the reorientation of the C2991 thiol group inside the loop stabilized by the interactions with L2980 (van der Waals interaction) and G2993 (hydrogen bond).

The effect of CaM/ARM210 on C2991 SH-group availability in phosphorylated R2474S RyR2 mutants.

The stabilizing agent ARM210 increased the C2991 thiol group ASA value in the phosphorylated C2474S mutant (PDB ID: 7UA1) in the closed conformation from 43.5 to 57.1 Å² (Supplementary information, Fig 9a). CaM binding in the same structure (PDB ID: 7UA3) had a smaller effect and increased ASA to 48.2 Å² (Supplementary information, Fig 9b). Interestingly, CaM binding to the C2474S mutant at the open

conformation (PDB ID: 7UA4) was able to reorient the C2991 thiol group outwards, restoring its ASA value from 29.4 to 55.7 Å².

Summary

In conclusion, we found that hRyR2s in open conformations were usually associated with a more exposed thiol group of C2991. The phosphorylation of hRyR2 also increased the C2991 thiol group's ASA value, making it more solvent accessible. The introduction of R2474S CPVT mutation was associated with lower numbers of ASA, especially in the open conformation. Finally, the binding of stabilizing agents such as CaM or ARM210 increased the ASA values for the C2991 thiol group. However, it should be noted that even the conformation with the least accessible C2991 (PDB ID: 7U9Z) had an ASA value for the thiol group of 29.4 Å², which is ~36% from the maximal number (RSA per residue of 27%). Thus, in all analyzed RyR2 conformations, C2991 appears to be accessible for oxidation, with higher accessibility in open and phosphorylated conformations. While the precise conformations of the P1-SPRY2 linker and C1078 are yet to be established, this cysteine might be solvent accessible in some linker configurations. We have added this analysis to the revised manuscript (the main text and the supplementary information file).

3. *RyRs are known to form higher-order interactions with neighboring RyRs, often in a regular, 2D-crystalline manner. The Cys2991 is close to a proposed inter-RyR interface. How are the authors certain that they are not looking at inter-RyR interactions? Or a combination of intra-RYR (but intersubunit) and inter-RyR disulfide bonds? The SDS-PAGE results cannot discriminate between these possibilities. Unless there is other evidence, this possibility should be discussed.*

Our data suggest that the individual mutations of C1078 and C2991 lead to a similar reduction in redox-mediated cross-linking of hRyR2 along with a similar decrease in redox-mediated Ca²⁺ leak. The simultaneous mutation of both cysteines does not additively increase these protective effects, suggesting an interchangeable role of these cysteines in the mechanism of protection. Intersubunit disulfide bond formation involving both cysteines provides a good explanation for these effects. The spatial proximity and surface accessibility of these two cysteines suggests the intersubunit disulfide bond formation within one channel is the most preferred mechanism for cross-linking. Available structural information indicates the localization of cross-linking cysteines at the inner surface of the RyR2 subunits¹⁴. Both the P1-SPRY2 linker harboring C1078 and the loop from the HD2 domain that includes C2991 are parts of the RyR2 inner intersubunit area and do not belong to the proposed channel-channel self-organization interfaces¹⁷. However, we do not specifically claim that identified C1078 and C2991 are the only cysteines capable of disulfide bond formation in RyR2. Additional cysteines might still be involved in disulfide formation in other sites of RyR2, including interactions between channels.

4a. *The mechanistic implication of an inter-subunit disulfide bond is only vaguely suggested. The authors did not capitalize on various available cryo-EM structures of both RyR1 and RyR2 in different states. As C1078 is in a flexible linker, invisible in most RyR cryo-EM structures, there is considerable 'slack' to allow this Cys to approach e.g. Cys2991. A straightforward mechanism would be that the distance between the two cysteines is too large to allow disulfide bond formation in a particular conformation of the RYR. I strongly urge the authors to take high-resolution structures of RyR2 in distinct conformations, and model in the flexible loop with Cys2991, seeing if it can physically reach Cys1078, while satisfying all geometrical parameters (bond lengths and angles, VDW clashes etc). If this is sterically possible in some, but not all states, then this immediately provides a mechanism. Another possibility is that the surface accessibility of the cysteines changes as a function of conformation, which also would reveal a mechanism.*

Available Cryo-EM structures¹⁴ provide important information about the orientation of C2991, whereas the exact conformation of C1078 on the flexible linker seems unclear due to limited resolution in this area.

However, we can still use those models to evaluate the possibility of the C1078-C2991 disulfide bond formation. To investigate possible conformations of the P1-SPRY2 linker and specifically C1078, we compared the cryo-EM models of the closed RyR2^{WT} at resolution 2.83 Å (PDB ID: 7UA5)¹⁴ and two structures of closed RyR2^{WT} and RyR2^{R176Q} (PDB IDs: 6WOV, 6WOU; resolutions 5.1 and 3.27 Å) published by another group¹⁸. Despite the CPVT mutation in the 6WOU model, both WT RyR2 and R176Q mutants share a similar P1-SPRY2 linker conformation. At the same time, the mutant model has a substantially better overall resolution (3.27 Å), which makes it an overall more reliable source of structural information. The position of another cross-linking cysteine (C2991) was derived from the 7UA5 model, which has a better resolution in this area.

The superposition (WinCoot 0.9.8.1, LSQ, C α) of the corresponding SPRY2 and P1 domains (Supplementary information, Fig 10a and b) showed that depending on the linker conformation, C1078 could be oriented towards C2991 (PDB IDs: 6WOV; 6WOU) or in the opposite direction (PDB ID: 7UA5). Both linker conformations seem to be geometrically allowed, given that all models were validated before the deposition to the PDB bank (Supplementary information, Fig 10b). Using 6WOU linker conformation as a template, we manually rebuilt the linker (WinCoot 0.9.8.1) to artificially introduce a disulfide bond between C2991 and C1078 (Supplementary information, Fig 10b). Obtained pseudo linker conformation was geometrically refined in WinCoot software to avoid the introduction of forbidden conformations (bonds, angles, rotamers). Based on this simple modeling and surface accessibility numbers, we believe C1078 may physically reach C2991 to form a disulfide bond (Fig 10c). At the same time, it appears that the P1-SPRY2 linker should be stabilized in a specific conformation to make the disulfide bond formation possible.

Therefore, we propose a model where an oxidant agent molecule (Diamide, GSSG) would bind C1078 and stabilize the linker in the favorable conformation for further interaction with C2991. In the next step, the C1078 would create a disulfide bond with C2991 releasing the reduced form of the oxidant molecule (hydrazine, GSH). Since the C2991 appears to be highly accessible in different gating states, the whole cycle of disulfide formation may include either close or open conformations of the channel or both. An additional structural investigation would be needed to reveal the conformations of C1078 and P1-SPRY2 linker favorable for the disulfide formation. We added this analysis to the revised manuscript (the main text and the supplementary information file).

4b. *As oxidation of RyRs goes hand-in-hand with disease-associated mutations, the authors could also take a look at disease mutant structures, as the Helical domain has been shown to undergo structural changes as a result of mutation in both RyR1 and RyR2. Finally, CaM binds to the helical domain, close to Cys2991, and cryo-EM studies on RyR1 and RyR2 bound to CaM have also shown conformational changes of the helical domain containing Cys2991. Especially since CaM is natively expressed in HEK293 cells, disulfide bond formation may thus affect CaM binding, therefore not allowing CaM modulation. Thus, the authors should also assess the sterical constraints to form disulfide bonds in CaM-bound conformations.*

Please see our answer to comment #2 for the complete structural analysis for CaM bound and CPVT mutation RyR2 structures (Supplementary information, Fig 8 and 9). In brief, the R2474S CPVT mutation in phosphorylated RyR2 may decrease the accessibility of the C2991 thiol group, especially in the open conformation. CaM binding in various states was associated with the increased ASA values for this group. At the same time, even the conformation with the least accessible C2991 has the SH-group RSA value of 36% and can be reached by solvent.

5a. *The authors started out with removing the N-terminal domains (A and B) of the RyR2. This seemed to have a big effect on the ability of RyR to form intersubunit disulfide bonds, but individual mutations in these domains failed to reproduce this effect. The authors concluded that removal of these domains resulted in allosteric effects. Removal of entire domains is indeed very drastic and it is even surprising that it leads to functional channels. In fact, a previous study deleting these domains has already shown that this results in different functional behavior (Liu et al , 2015, J.Biol.Chem. 290, 7736-7743). Although already referenced, the authors can also particularly use this reference to strengthen the argument that deleting these domains causes allosteric changes.*

The functional characterization of the AB N-terminal truncated RyR2 was not the goal of this study. Our primary focus was to identify cross-linking cysteines and to evaluate the functional significance of this post-translational modification. However, we noticed that the truncated RyR2 mutant is characterized by the decreased ability to generate spontaneous Ca²⁺ waves, suggesting a more inhibited RyR2. We will update our manuscript with this information and the reference of the previous study¹⁹.

5b. *In regards to the individual cysteines, the authors should have analyzed the precise environment of these, as they would have seen that they are not available for forming intersubunit interactions. Although they are somewhat 'close' on the scale of the full RyR2, they are in fact quite far from any Cys in a neighboring subunit, and there is quite some folded protein in between them. The result where the point mutations in these N-terminal domains have no effect on intersubunit disulfide bond formation is thus trivial and expected. The authors could still keep the corresponding data, but rather as a negative control, establishing the validity of the experimental approach.*

Despite the low solvent accessibility of the cysteine residues in AB domains of the RyR2 N-terminal, we were intrigued by the ability of the truncation mutant to prevent intersubunit cross-linking of RyR2. It has been reported that some cysteines from this region may become partially exposed to the solvent, such as C36 and C158²⁰. Thus, we validated the corresponding cysteine mutants for their ability to protect against channel intersubunit cross-linking. Based on the negative results of these experiments, we conclude that the truncation of the N-terminal domains prevents intersubunit cross-linking of RyR2 via allosteric changes to the channel's structure.

REFERENCES

1. Nikolaienko, R. *et al.* The functional significance of redox-mediated intersubunit cross-linking in regulation of human type 2 ryanodine receptor. *Redox Biol.* **37**, 101729 (2020).
2. Dulhunty, A., Haarmann, C., Green, D. & Hart, J. How many cysteine residues regulate ryanodine receptor channel activity? *Antioxid. Redox Signal.* **2**, 27–34 (2000).
3. Eager, K. R. & Dulhunty, A. F. Activation of the cardiac ryanodine receptor by sulfhydryl oxidation is modified by Mg²⁺ and ATP. *J. Membr. Biol.* **163**, 9–18 (1998).
4. Eager, K. R. & Dulhunty, A. F. Cardiac ryanodine receptor activity is altered by oxidizing reagents in either the luminal or cytoplasmic solution. *J. Membr. Biol.* **167**, 205–14 (1999).
5. Xu, L. Activation of the Cardiac Calcium Release Channel (Ryanodine Receptor) by Poly-S-Nitrosylation. *Science (80-.)*. **279**, 234–237 (1998).
6. Rousseau, E. & Meissner, G. Single cardiac sarcoplasmic reticulum Ca²⁺-release channel: activation by caffeine. *Am. J. Physiol.* **256**, H328-33 (1989).
7. Sawyer, D. B. *et al.* Role of oxidative stress in myocardial hypertrophy and failure. *J. Mol. Cell.*

- Cardiol.* **34**, 379–388 (2002).
8. Belevych, A. E. *et al.* Redox modification of ryanodine receptors underlies calcium alternans in a canine model of sudden cardiac death. *Cardiovasc. Res.* **84**, 387–395 (2009).
 9. Bovo, E., Mazurek, S. R. & Zima, A. V. The role of RyR2 oxidation in the blunted frequency-dependent facilitation of Ca²⁺ transient amplitude in rabbit failing myocytes. *Pflugers Arch.* **470**, 959–968 (2018).
 10. Bovo, E., Mazurek, S. R. & Zima, A. V. Oxidation of ryanodine receptor after ischemia-reperfusion increases propensity of Ca²⁺ waves during β -adrenergic receptor stimulation. *Am. J. Physiol. Circ. Physiol.* **315**, H1032–H1040 (2018).
 11. Liu, T., Yang, N., Sidor, A. & O'Rourke, B. MCU Overexpression Rescues Inotropy and Reverses Heart Failure by Reducing SR Ca²⁺ Leak. *Circ. Res.* **128**, 1191–1204 (2021).
 12. Terentyev, D. *et al.* Redox modification of ryanodine receptors contributes to sarcoplasmic reticulum Ca²⁺ leak in chronic heart failure. *Circ. Res.* **103**, 1466–72 (2008).
 13. Zaidi, N. F., Lagenaur, C. F., Abramson, J. J., Pessah, I. & Salama, G. Reactive disulfides trigger Ca²⁺ release from sarcoplasmic reticulum via an oxidation reaction. *J. Biol. Chem.* **264**, 21725–36 (1989).
 14. Miotto, M. C. *et al.* Structural analyses of human ryanodine receptor type 2 channels reveal the mechanisms for sudden cardiac death and treatment. *Sci. Adv.* **8**, eabo1272 (2022).
 15. Samanta, U., Bahadur, R. P. & Chakrabarti, P. Quantifying the accessible surface area of protein residues in their local environment. *Protein Eng.* **15**, 659–67 (2002).
 16. Savojardo, C., Manfredi, M., Martelli, P. L. & Casadio, R. Solvent Accessibility of Residues Undergoing Pathogenic Variations in Humans: From Protein Structures to Protein Sequences. *Front. Mol. Biosci.* **7**, 1–9 (2021).
 17. Cabra, V., Murayama, T. & Samsó, M. Ultrastructural Analysis of Self-Associated RyR2s. *Biophys. J.* **110**, 2651–2662 (2016).
 18. Iyer, K. A. *et al.* Structural mechanism of two gain-of-function cardiac and skeletal RyR mutations at an equivalent site by cryo-EM. *Sci. Adv.* **6**, 1–13 (2020).
 19. Liu, Y. *et al.* Roles of the NH₂-terminal domains of cardiac ryanodine receptor in Ca²⁺ release activation and termination. *J. Biol. Chem.* **290**, 7736–7746 (2015).
 20. Lobo, P. A. & Van Petegem, F. Crystal Structures of the N-Terminal Domains of Cardiac and Skeletal Muscle Ryanodine Receptors: Insights into Disease Mutations. *Structure* **17**, 1505–1514 (2009).

Reviewers' Comments:

Reviewer #1:

Remarks to the Author:

I thank the authors for providing additional data regarding store of SR/ER filling with SERCA inhibitors. The selective cross-linking of C1078 and C2991 remain heuristic and largely unproven, at least directly. The use of mass spectrometry approaches could directly prove or disprove the purported cross linking of these thiols and directly address specificity of the diamine approach used.

Reviewer #2:

Remarks to the Author:

The authors have done a good job at addressing previous concerns and recommendations. This reviewer does worry about the fact that the cross-linked RyR2 seems to be stuck in the wells of the SDS-PAGE/Western blots. This means that one cannot discriminate between regular tetramers (implying only intra-RyR2 crosslinks) and higher-order aggregates (possibly including multiple RyR2s linked together). Although the authors make the argument that intra-RyR2 is more likely, this statement would be much stronger with experimental proof. This could be done via a size exclusion column that shows the cross-linked RyR2 runs at the expected size. They could run detergent-solubilized cell membrane pellets on a gel filtration column with high-MW separation (e.g. Superose 6), and use western blot to detect where the protein runs. Cross-linked RyR2 should run at the same elution volume as non-cross-linked if all of the sample is tetrameric.

I realize this was not a request before, as the previous version of the manuscript did not indicate cross-linked RYR2 to stay in the wells. I had indicated this as a possible concern to be addressed if indeed RyR2 is in the wells, which now appears to be the case. I strongly recommend the authors to still do the gel filtration chromatography, which should take minimal time to complete.

Minor comment:

The cysteine accessibilities are now shown, but these are the full relative accessibilities. Please also indicate the side chain accessibility, as the accessibility of the main chain atoms is less important.

Nature Communications manuscript NCOMMS-22-45497: “Cysteines 1078 and 2991 cross-linking plays a critical role in redox regulation of cardiac RyR”.

We are thankful for the thoughtful suggestions and constructive criticisms provided by the reviewers. We have revised the manuscript to address all reviewer suggestions. A point-by-point response to each comment is provided below.

Reviewer #1:

The selective cross-linking of C1078 and C2991 remain heuristic and largely unproven, at least directly. The use of mass spectrometry approaches could directly prove or disprove the purported cross-linking of these thiols and directly address specificity of the diamine approach used.

We appreciate the reviewer’s suggestion to perform the mass-spectrometry experiment to validate the specificity of cross-linking between cysteines 1078 and 2991. Although the mass spectrometry approach may be helpful in the screening for the redox-sensitive cysteines in RyR2, it is still challenging to reliably identify residues involved in the intersubunit cross-linking. Post-translational modifications related to cysteine oxidation may involve disulfide bridge formation within one channel subunit and mixed disulfides with non-RyR2 molecules. Previous mass-spectrometry studies intended to identify reactive cysteines in RyR1 resulted in discovering the subset of cysteines available for oxidation^{1,2}. Some of those cysteines (C36, C3635, C2326, C2363) have been suggested to be involved in cross-linking², but their binding partners have not been identified, and their direct involvement in cross-linking has not been proven directly. Thus, the mass spectrometry approach might be a good tool for the initial candidate screening, but not for validation. Nevertheless, we have performed additional experiments to validate the cross-linking cysteines using the mass-spectrometry approach. Although we had a RyR2 peptide coverage of 30% with 144 different fragments, peptides with the cysteine residues of interest could not be detected (Table 1; examples of several RyR2 fragments). The 1056 peptide was closest to the cross-linking C1078. Furthermore, the subset of obtained peptides mostly did not contain regions with cysteine residues.

Table 1

sp|Q92736|RyR2_HUMAN Ryanodine receptor 2 OS=Homo sapiens OX=9606
GN=RyR2 PE=1 SV=3

Start Position on Q92736	Annotated Sequence	MS
1056	[R]. TLLGYGNLEAPDQDHAAR .[A]	HCD_IT
1134	[R].AFAFDGFK.[A]	HCD_OT
2949	[R].GKGHEHFPYEQEIK.[F]	HCD_OT
3145	[K].SIYVER.[Q]	HCD_IT

-AVR**TLLGYGNLEAPDQDHAAR**AEVCSGT-
↑
C1078

At the same time, our approach (site-directed mutagenesis of the rationally-selected cysteines based on analysis of novel cryo-EM models) provides strong direct evidence for the selective intersubunit cross-linking between cysteines C1078 and C2991. The analysis of the recently published RyR2 Cryo-EM models³ identified C1078 and C2991 as two cysteines available for oxidation and geometrically capable of disulfide bond formation. Biochemical and functional experiments in HEK cells further confirmed this prediction. To increase the specificity of RyR2 oxidation, we replaced the treatment of cells with diamide with treatment of oxidized glutathione (GSSG; Fig. 1). The GSSG treatment resulted in nearly complete cross-linking of RyR2, whereas the RyR2^{C1078S/C2991S} mutant preserved its monomeric form. These results suggest that RyR2 cross-linking mediated by cysteines 1078 and 2991 is specific and likely to occur during pathological conditions associated with decreased GSH/GSSG ratios.

Figure 1

The mutation of cysteines 1078 and 2991 to serines prevents disulfide cross-linking of hRyR2 during oxidation by GSSG. **a.** Representative western blot of the GFP-labeled hRyR2 and hRyR2^{C1078S/C2991S} treated with increasing GSSG concentrations in permeabilized HEK293 cells. **b.** the effect of the increasing GSSG concentrations on the level of RyR2 monomeric form for hRyR2^{WT} and hRyR2^{C1078S/C2991S} (n=3), data are shown as means ± SE and were analyzed using a two-sample t-test (**p < 0.01, ***p < 0.001).

Reviewer #2:

This reviewer does worry about the fact that the cross-linked RyR2 seems to be stuck in the wells of the SDS-PAGE/Western blots. This means that one cannot discriminate between regular tetramers (implying only intra-RyR2 cross-links) and higher-order aggregates (possibly including multiple RyR2s linked together). Although the authors make the argument that intra-RyR2 is more likely, this statement would be much stronger with experimental proof. This could be done via a size exclusion column that shows the cross-linked RyR2 runs at the expected size. They could run detergent-solubilized cell membrane pellets on a gel filtration column with high-MW separation (e.g. Superose 6), and use western blot to detect where the protein runs. Cross-linked RyR2 should run at the same elution volume as non-cross-linked if all of the sample is tetrameric.

As the reviewer pointed out, some representative images of our western blots contain the cross-linked fraction of RyR2 stuck in the well. This fact may raise a concern that cross-linked RyR2 is formed by the channel-channel aggregation rather than specific cross-linking between channel subunits. However, in our opinion, this is mainly associated with the high molecular weight of the cross-linked channel, which exceeds 1 MDa for the dimer and can reach even higher values for trimers and tetramers. When we ran western blots for WT and RyR2 mutants for extended time period, we observed distinct bands of high oligomeric forms of RyR2 that traveled below loading wells (Fig. 2). We usually observe three well-defined RyR2

Figure 2

The cross-linked hRyR2 forms a distinct pattern of oligomeric bands not observed in hRyR2 cysteine mutants. Representative western blots of the unlabeled hRyR2, hRyR2^{C1078S}, and hRyR2^{C2991S} treated with increasing diamide concentrations. The position of 1024 kDa band was verified by the IgM pentameric band from the NativeMark™ protein standard (Thermo Fisher Scientific). The bottoms of the protein gel wells are marked with black arrows.

bands over 1 MDa, in contrast to one smeared band typical for non-specific aggregates. Notably, no oligomeric bands were formed for the single cysteine mutants, suggesting the specific mechanism for RyR2 cross-linking involves cysteines 1078 and 2991 (Fig. 2). As we mentioned in our previous response, these cysteines are unlikely to be involved in channel-channel interactions due to their structural localization.

We are thankful for reviewer's suggestion to perform the analytical gel filtration experiment to rule out the channel-channel aggregation of the cross-linked RyR2. While the size exclusion chromatography of oxidized RyR2 appears to be an excellent way to demonstrate the preferential formation of intersubunit RyR2 oligomers rather than high-order aggregates, our laboratory lacks corresponding capacities for purification, solubilization and further analytical chromatography of RyR2. To strengthen our conclusion, we performed fluorescence recovery after photobleaching (FRAP) experiments in HEK293 cells expressing GFP-RyR2. FRAP of membrane proteins with a fluorescent tag is commonly used to estimate changes in the lateral diffusion rates. The lateral diffusion is negatively correlated with a size of membrane proteins. Therefore, if RyR2 oxidation causes the channel-channel cross-linking, it should significantly reduce the lateral diffusion rate and FRAP. We used 5 mM GSSG to oxidize RyR2. This [GSSG] caused a similar effect on RyR2 cross-linking in permeabilized cells as 50 μ M diamide in intact cells, but didn't cause cross-linking in RyR2^{C1078S/C2991S} (Fig. 1). The relative FRAP values were measured in control cells and cells treated with 5 mM GSSG for 20 min. The treatment cells with GSSG did not change the FRAP rate of GFP-RyR2 (Fig. 3), suggesting the absence of channel-channel aggregation. Thus, our data indicates that intersubunit RyR2 cross-linking, but not channel-channel aggregation, appears to be the preferred mechanism for the redox-dependent formation of higher oligomeric forms of RyR2.

Minor comment:

The cysteine accessibilities are now shown, but these are the full relative accessibilities. Please also indicate the side chain accessibility, as the accessibility of the main chain atoms is less important.

All ASA values in our Figures are the absolute area values (\AA^2) calculated for cysteine thiol groups. We will make sure to clarify this in all Figure legends. Also, please refer to the supplementary Table 1, which contains absolute and relative ASA values for both thiol groups and entire cysteine residues.

References

1. Voss, A. A., Lango, J., Ernst-Russell, M., Morin, D. & Pessah, I. N. Identification of hyperreactive cysteines within ryanodine receptor type 1 by mass spectrometry. *J. Biol. Chem.* **279**, 34514–34520 (2004).
2. Aracena-Parks, P. *et al.* Identification of cysteines involved in S-nitrosylation, S-glutathionylation, and oxidation to disulfides in ryanodine receptor type 1. *J. Biol. Chem.* **281**, 40354–40368 (2006).
3. Miotto, M. C. *et al.* Structural analyses of human ryanodine receptor type 2 channels reveal the mechanisms for sudden cardiac death and treatment. *Sci. Adv.* **8**, eabo1272 (2022).

Reviewers' Comments:

Reviewer #1:

Remarks to the Author:

Thank you for attempting to substantiate your conclusions with mass spectroscopy. I realize the authors are not experts in or familiar with this area and the techniques and databases that accommodate the search algorithms to identify cross linked partners.

Reviewer #2:

Remarks to the Author:

The authors have address all of my previous concerns.